# Neurotoxic amyloidogenic peptides in the proteome of SARS-COV2: potential implications for neurological symptoms in COVID-19

Mirren Charnley[1,2], Saba Islam[3], Guneet K. Bindra[3], Jeremy Engwirda[3], Julian Ratcliffe[4], Jiangtao Zhou[5], Raffaele Mezzenga[5], Mark D. Hulett[3], Kyunghoon Han[6], Joshua T. Berryman[6✉] & Nicholas P. Reynolds[3✉]

COVID-19 is primarily known as a respiratory disease caused by SARS-CoV-2. However, neurological symptoms such as memory loss, sensory confusion, severe headaches, and even stroke are reported in up to 30% of cases and can persist even after the infection is over (long COVID). These neurological symptoms are thought to be produced by the virus infecting the central nervous system, however we don't understand the molecular mechanisms triggering them. The neurological effects of COVID-19 share similarities to neurodegenerative diseases in which the presence of cytotoxic aggregated amyloid protein or peptides is a common feature. Following the hypothesis that some neurological symptoms of COVID-19 may also follow an amyloid etiology we identified two peptides from the SARS-CoV-2 proteome that self-assemble into amyloid assemblies. Furthermore, these amyloids were shown to be highly toxic to neuronal cells. We suggest that cytotoxic aggregates of SARS-CoV-2 proteins may trigger neurological symptoms in COVID-19.

[1] Centre for Optical Sciences and Department of Health Sciences and Biostatistics, Swinburne University of Technology, Hawthorn, VIC 3122, Australia. [2] Immune Signalling Laboratory, Peter MacCallum Cancer Centre, Parkville, VIC 3000, Australia. [3] Department of Biochemistry & Chemistry, La Trobe Institute for Molecular Science, La Trobe University, Bundoora, VIC 3086, Australia. [4] La Trobe University Bioimaging Platform, Bundoora 3086 VIC, Australia. [5] Department of Health Sciences & Technology, ETH Zurich, Schmelzbergstrasse 9, LFO, E23, 8092 Zurich, Switzerland. [6] Department of Physics and Materials Science, Faculty of Science, Technology and Medicine, University of Luxembourg, 162a Avenue de la Faïencerie, Esch-sur-Alzette L-1511, Luxembourg. ✉email: josh.berryman@uni.lu; nicholas.reynolds@latrobe.edu.au

The disease caused by viral infection with severe acute respiratory syndrome (SARS)-COV-2 is known as COVID-19 and whilst predominantly a respiratory disease affecting the lungs it has a remarkably diverse array of symptoms. These include a range of moderate to severe neurological symptoms reported in as many as 30% of patients, which can persist for up to 6 months after infection[1]. These symptoms include memory loss, sensory confusion (e.g., previously pleasant smells become fixed as unpleasant), cognitive and psychiatric issues, severe headaches, brain inflammation and haemorrhagic stroke[1–5]. COVID-19-related anosmia and phantosmia have been shown to correlate with a persistence of virus in the olfactory mucosa and in the olfactory bulb of the brain, and with persistent inflammation; however, negative evidence for continuing viral replication has also been shown for long-term anosmia[6]. Furthermore, there is evidence that SARS-CoV-2 is neuroinvasive with either the full virus[7,8] or viral proteins[8] being found in the CNS of mouse models and the post-mortem brain tissue of COVID-19 patients. Whilst the neuroinvasiveness of SARS-CoV-2 is apparent the molecular origin of the associated neurological symptoms is as yet unknown, although they are similar to hallmarks of amyloid-related neurodegenerative diseases such as Alzheimer's (AD)[9,10], and Parkinson's[11]. For instance, impaired olfactory identification ability and mild cognitive impairment have also been reported in the early stages of AD and prodromal AD[12].

A number of in vitro studies have shown that proteins from SARS-CoV-2 can detrimentally affect a variety of cell types including kidney, liver and immune cells[13,14]. Furthermore, experiments on brain organoids show that SARS-CoV-2 can infect neuronal cells resulting in cell death[15]. Combined these papers point to a potential cytotoxic cause of neurological symptoms in COVID-19.

Proteins from the Zika virus[16] and also the coronavirus responsible for the SARS outbreak in 2003 (SARS-COV-1)[17] have been shown to contain sequences that have a strong tendency to form amyloid assemblies. As the proteome of SARS-CoV-1 and SARS-CoV-2 possess many similarities[18], we propose amyloid nanofibrils formed from proteins in SARS-CoV-2 may be implicated in the neurological symptoms in COVID-19. Therefore, amyloid-forming proteins from the SARS-CoV-2 virus in the CNS of COVID-19 infected patients could have similar cytotoxic and inflammatory functions to amyloid assemblies that are the molecular hallmarks of amyloid-related neurodegenerative diseases such as AD (Aβ, Tau) and Parkinson's (α-synuclein). The worst-case scenario given the present observations is that of the progressive neurological amyloid disease being triggered by COVID-19. To the authors' knowledge, there has so far been no documented example of this; however, it has been noted that up-regulation of Serum amyloid A protein driven by inflammation in COVID-19 seems like a probable trigger for the systemic (non-neurological) amyloid disease AA amyloidosis[19], which is already known to be a concomitant of inflammatory disease in general.

If the proteome of SARS-COV-2 does contain amyloid-forming sequences, this raises the question, what is their function? It is known that viral genomes evolve rapidly and are highly constrained by size; therefore, every component is typically functional either to help the virus replicate or to impede the host immune system. To this end, there are several potential roles for amyloid assemblies in pathogens generally[20] and specifically in coronaviruses such as SARS-CoV-2. The simplest is that amyloid is an inflammatory stimulus[21], and proinflammatory cytokines can up-regulate the expression of the spike protein receptor ACE-2 such that intercellular transmissibility of SARS-CoV-2 is increased. Alternatively Tayeb-Fligelman et al.[22] found that the nucleocapsid protein in SARS-CoV-2, which is responsible for packaging RNA into the virion, contains a number of highly amyloidogenic short peptide sequences within its intrinsically disordered regions[22]. It has been shown that the self-assembly of these peptides is enhanced in the presence of viral RNA, during liquid–liquid phase separation (LLPS is an important stage in the viral replication cycle)[23,24]. These findings suggest amyloids may play an important role in RNA binding and packaging during the viral replication cycle. Finally, it is also possible that amyloid assemblies in coronaviruses might have a role in inhibiting the action of the host antiviral response similar to a discussed role for amyloid in other viruses. Pham et al.[25] observed that amyloid aggregates from murine cytomegalovirus can interfere with RIPK3 kinase activation and potentially inhibit its antiviral immune signalling capabilities.

In this study, we choose to focus on a selection of proteins from the SARS-CoV-2 proteome known as the open reading frames (ORFs). These ORF proteins were chosen as they have no obvious roles in viral replication[26], perhaps freeing them up to have yet uncharacterised roles in disrupting the host antiviral responses. By sequence and length, they appear to be largely unstructured, making them good candidates for amyloid formation in vivo. We performed a bioinformatic screening of the ORF proteins to look for potential amyloidogenic peptide sequences. This analysis was used to select two sub-sequences, one each from ORF6 and ORF10, for synthesis. The synthesised peptides were both found to rapidly self-assemble into amyloid assemblies with a variety of polymorphic morphologies. Cytotoxicity assays on neuronal cell lines showed these peptide assemblies to be highly toxic at concentrations as low as 0.0005% (0.04 mg mL$^{-1}$).

Since commencing this work, others have found that ORF6 is the most cytotoxic single protein of the SARS-CoV-2 proteome, showing localisation to membranes when overexpressed in human and primate immune cell lines[13]. In contrast, ORF10 has been reported as an unimportant gene with very low expression and no essential role in virus replication[26]; however, the functions of immune suppression or inflammation promotion via amyloid formation would be non-essential, if present, and should not necessarily require transcription in large volumes, making ORF10 an intriguing second candidate for the present study. It is also interesting that ORF10 and ORF8 are the only two coded proteins present in SARS-CoV-2 which do not have a homologue in SARS-CoV-1[27], perhaps suggesting a unique amyloid etiology for COVID-19. While long-term consequences from SARS-CoV-1 infection were severe, including tiredness, depression, and impaired respiration, few or zero unequivocally neurological post-viral symptoms were recorded from the (admittedly quite small) set of documented cases[28].

## Results and discussion

**Amyloid aggregation prediction algorithms identified two short peptides from ORF6 and ORF10 that are likely to form amyloids.** Figure 1 shows selected output from bioinformatics tools applied to predict the amyloidogenicity of peptide sequences within larger polypeptides. Application of the ZIPPER tool to ORF6 provides more than ten choices of six-residue windows of the sequence predicted to be highly amyloidogenic (Fig. 1a), while ORF10 shows only three such highly amyloidogenic sequence windows (Fig. 1b). To narrow down our search for candidate peptides we also used the TANGO algorithm, for ORF6 there are two regions that are predicted to be highly aggregation prone, $I_{14}$LLIIMR and $D_{30}$YIINLIIKNL. The region $I_{14}$-$R_{20}$ overlaps almost perfectly with the hexapeptide $I_{14}$LLIIM identified by ZIPPER. The region 30–40 also contains multiple hits in ZIPPER, but as this study was limited to two candidate peptides we chose ILLIIM as our first candidate as it closely resembles the sequence

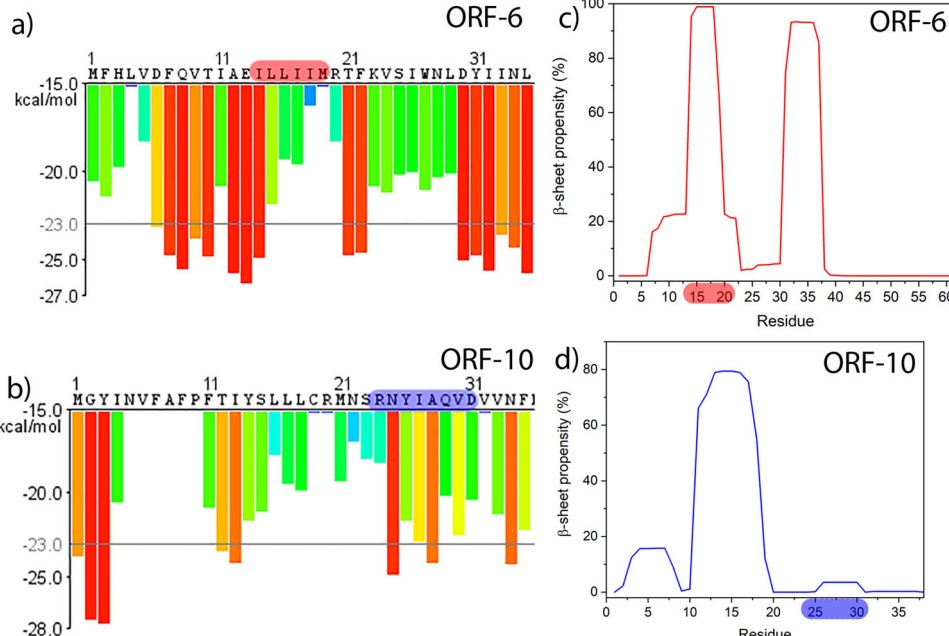

**Fig. 1 Output from amyloid assembly prediction software for SARS-CoV-2 ORF6 and ORF10 sequences. a, b** Outputs from amyloid predicting algorithm ZIPPER identifying hexapeptide fragments predicted to form steric zippers nucleating the assembly of amyloid fibrils. Hexapeptides with Rosetta energies below −23 kcal mol⁻¹ are predicted to be highly amyloidogenic and are shown with red bars. **c, d** Outputs from amyloid predicting algorithm TANGO indicating sequence regions of high propensity to form β-sheets.

ILQINS from Hen Egg White Lysozyme that has previously been seen to be highly amyloidogenic (the mutation TFQINS in human lysozyme is disease-linked)[29–31]. Looking now at the TANGO plots for ORF10 the main aggregation-prone sequence is residues $F_{11}TIYSLLLC$, although there are no high stability hexapeptides in this sequence predicted by ZIPPER. The octapeptide $R_{24}NYIAQVD$ was chosen due to its zwitterionic residue pair R-D which should strongly enhance interpeptide association, despite being too far apart in the sequence to trigger the highly local bioinformatics algorithms. Encouragingly ZIPPER also predicts the hexapeptide NYIAQV contained within RNYIAQVD to be highly amyloidogenic. Based on the outputs from ZIPPER and TANGO and also on the experience in making and studying amyloid, we selected RNYIAQVD and ILLIIM to be synthesised and their amyloid-forming capability investigated.

**Nanoscale imaging reveals both peptide sequences self-assemble into polymorphic amyloid assemblies.** Atomic force microscopy (AFM) imaging of the two peptide assemblies revealed that both peptides assembled at 37 °C almost immediately at 1 mg mL⁻¹ (Supplementary Figs. 1 and 2) into a highly polymorphic mixture of nanofibrous and crystalline structures (Supplementary Fig. 3). For both peptides, the dominant polymorph was needle-like crystalline assemblies as seen in Fig. 2. In an attempt to ensure any observed polymorphism was not due to a heterogeneous mixture of insoluble peptide seeds we added warm PBS (90 °C) to the lyophilised peptides, and maintained this elevated temperature for at least 3 h in order dissolve as much of the monomeric peptide as possible. Self-assembly was subsequently initiated by slowly reducing the temperature, using a previously developed protocol[32]. This method produced less polymorphism resulting in the needle-like crystalline polymorph being overwhelmingly dominant, however a number of twisted fibrillar polymorphs were still present for RNYIAQVD (Fig. 2i and Supplementary Fig. 4a). To facilitate repeatable quantitative analysis of the biochemical and biophysical properties of the

assemblies we used the slow cooling assembly method for all further experiments.

AFM and transmission electron microscopy (TEM) imaging of assemblies formed at either 1 or 5 mg mL⁻¹ for 24 h revealed that assemblies from both peptides tend to stack on top of each other forming multi-laminar structures (Fig. 2a–d and Supplementary Figs. 5 and 6). Evidence of lateral assembly of the needles was also observed but this appears to happen more frequently in the ILLIIM assemblies (Fig. 2g) compared to RNYIAQVD. ILLIIM tends to form very large (2–3 μm in width) multi-laminar crystalline assemblies (Fig. 2g), whereas RNYIAQVD predominantly forms long linear needle-like structures. The apparent lower tendency of RNYIAQVD to form large two-dimensional lateral assemblies can be explained by the polymorphism seen in this peptide, which does not promote translational symmetry (i.e., extended crystals). Figure 2i and Supplementary Fig. 4a both clearly show that in addition to the flat needle-like crystals seen elsewhere, RNYIAQVD can also form non-planar partially twisted fibrillar assemblies. This polymorphism observed in RNYIAQVD assemblies may reduce the ability of the crystals to laterally associate and stack into multi-laminar species, simply because of a mismatch in planarity between two adjacent assemblies, the molecular basis for this polymorphism is briefly discussed in the next section of the manuscript. AFM was further used to investigate the height of the individual assemblies of both ILLIIM and RNYIAQVD. Figure 2a, b shows a line section through a multi-laminar RNYIAQVD assembly with two distinct layers with a step height of 5.5 nm between each layer. Similarly for ILLIIM (Fig. 2c and Supplementary Fig. 5), we see multi-laminar stacking with individual step heights that vary between 4 and 12 nm. Turning to RNYIAQVD we see single crystals with step heights varying between 5 and 20 nm (Supplementary Fig. 6). Together this heterogeneous distribution of crystal heights provides further evidence for the polymorphic nature of both the ILLIIM and RNYIAQVD assemblies.

Quantitative analysis of the distribution of assembly heights and contour lengths was performed using the freely available

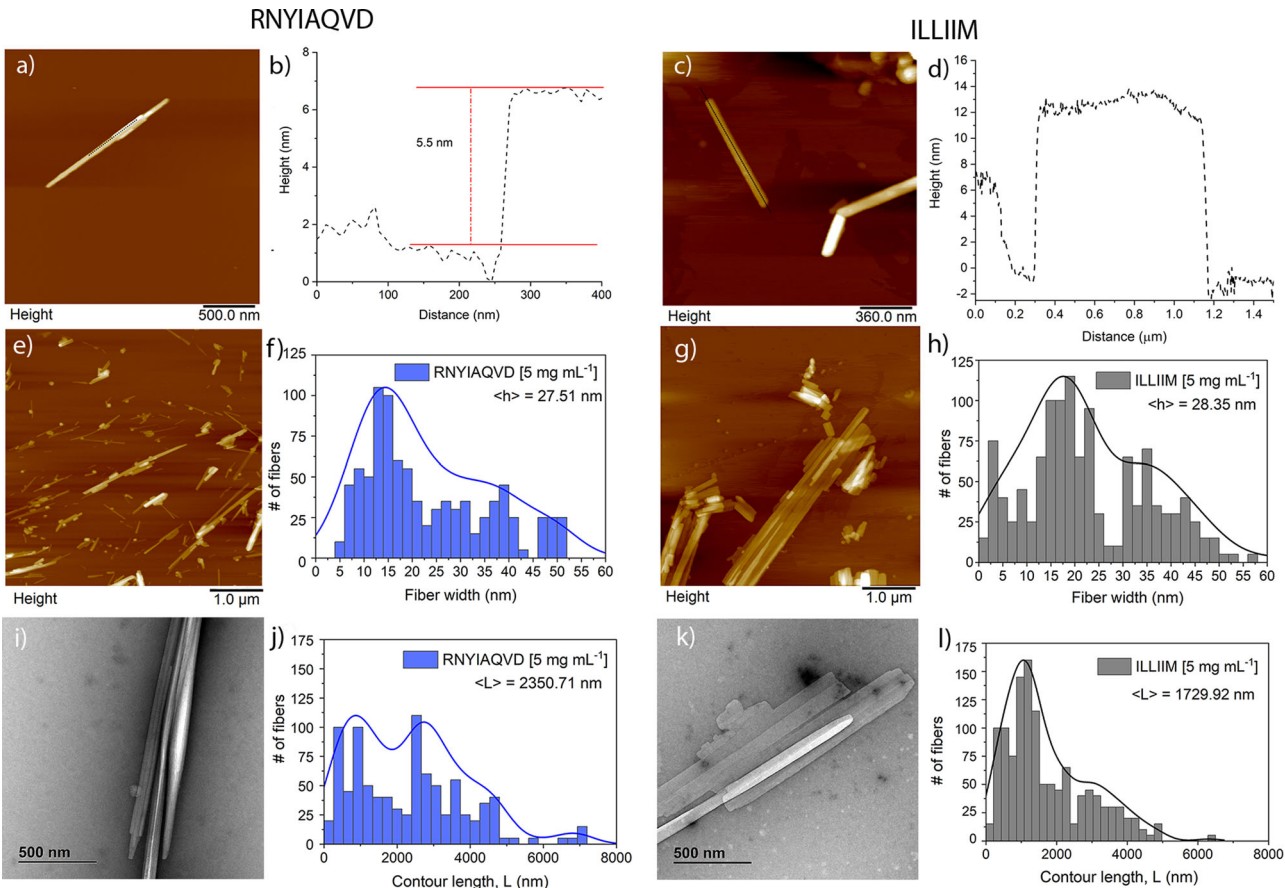

**Fig. 2 Atomic force and transmission electron microscopy images of peptide assemblies at 5 mg mL$^{-1}$ incubated for 24 h. a–d** AFM images and linescans showing variation in assembly width and assembly stacking for both peptides, all AFM images are taken at 512 × 512 scan lines × pixels, at a scanning speed of 0.5 Hz displayed at a z-height of 40 nm using a black–golden–white colour gradient. **e–h** additional lower magnification AFM images of both assemblies and corresponding fibre width quantitative analysis. **i–l** TEM images of the two peptide assemblies, and contour length analysis of the two assemblies (from AFM data). All statistical analysis was performed on datasets including at least 900 assemblies, all histogram fitting was performed in Origin Pro using the Kernel Smooth fitting parameters with Scott Bandwith, scaled to 100% of the maximum intensity of the histogram. All images are representative images taken from three independent experiments. Additional examples of TEM and AFM images from other independent experiments can be found in the Supplementary information (Supplementary Figs. 3–6).

software tool FiberApp[33]. The analysis of assembly height distribution was taken from the z-axis (height) of the AFM images, both peptide assemblies show a heterogeneous distribution of fibril heights due to the previously observed tendency of both assemblies to form polymorphic multi-laminar stacks (Fig. 2f, h). Analysis of the distribution of the contour length of the two assemblies showed a biphasic distribution of lengths for both fibrils with two broad sub-populations centred around 1 and 3 μm (Fig. 2j, l). The sub-population at 3 μm was seen to be much larger for the RNYIAQVD peptide (Fig. 2j) compared to the ILLIIM (Fig. 2l). This population of longer fibrils correlates with the observation from Fig. 2 that for RNYIAQVD longer, thinner assemblies are favoured (self-assembly via fibril extension) over the wider shorter assemblies more commonly seen for ILLIIM (assembly via lateral association of protofilaments). Analysis of the persistence length of the fibrils (Supplementary Fig. 7) showed that whilst both peptides formed very straight linear assemblies, the persistence length of RNYIAQVD ($\lambda = 41.92\,\mu m$) is greater than that of ILLIIM ($\lambda = 31.96\,\mu m$).

To further investigate the polymorphic nature of the assembly of these peptides, we investigated the structures formed from 1:1 mixtures of the two peptides. Interestingly when mixed prior to assembly the peptides form a wide range of polymorphic structures exceeding that of either peptide assembled individually.

Supplementary Fig. 8 shows a selection of some of the polymorphs formed, especially interesting are the large flat structures with well-defined edges that seem almost to interlock (Supplementary Fig. 8c). Such well-ordered 2D crystals were never observed for either peptide individually, and provide clear evidence of co-crystallisation. At this stage we have no evidence for the biological relevance of this co-crystallisation; however, as the ORF proteins from which these peptides are identified are themselves very small proteins (ORF6 is 61 amino acids in length), it is feasible that these small proteins may undergo similar co-crystallisation during their viral replication cycle facilitating an as yet unknown biological function.

**X-ray scattering, spectroscopic characterisation, fluorescent microscopy and molecular modelling confirm the amyloid nature of the assemblies.** Figure 3a shows the radially averaged 1D small-angle X-ray scattering (SAXS) plots for ILLIIM and RNYIAQVD at the lower concentration studied (at the higher concentration, sedimentation made recording X-ray scattering spectra impossible). In the central part of the scattering curve, the ILLIIM assemblies produced a slope with a $q^{-2}$ dependence which is consistent with the form factor of an infinite 2D surface[30], and is most likely arising due to the broader lateral

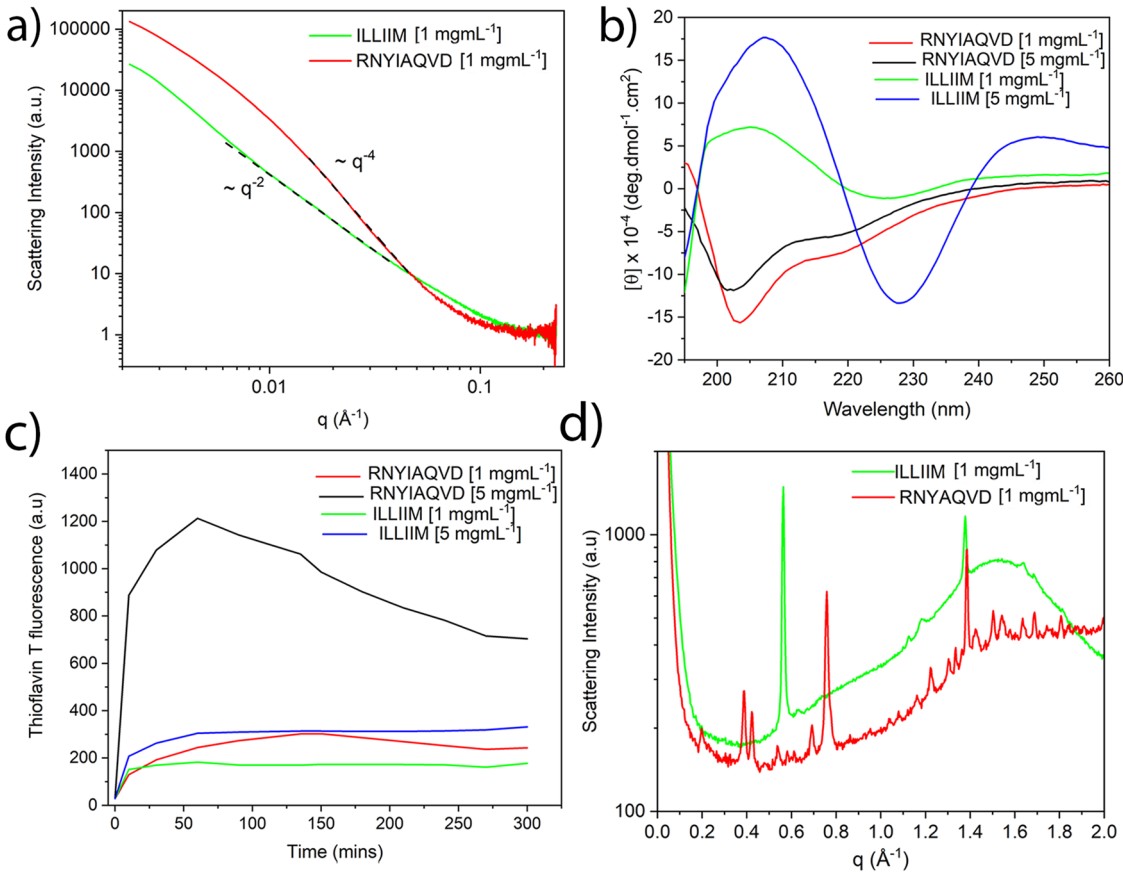

**Fig. 3 Spectroscopic analysis of the secondary structure of the peptide assemblies. a** 1D SAXS plots showing $q^{-2}$ dependence for ILLIIM (flat ribbon) and $q^{-4}$ dependence for RNYIAQVD, **b** circular dichroism spectroscopy of the two peptide assemblies and **c** Thioflavin T spectroscopy to quantify the amount of amyloid formation of the two peptide assemblies. **d** WAXS spectra of ILLIIM and RNYIAQVD assemblies. SAXS and WAXS plots are shown as background subtracted against PBS in the same quartz capillary and averaged from approximately 15 recorded spectra.

dimensions observed by AFM and TEM for ILLIIM compared to RNYIAQVD. RNYIAQVD, however, displays a $q^{-4}$ dependence in the central part of the scattering curve, appearing more towards the high-$q$ limit. Porod's law indicates that $q^{-4}$ scaling (at high $q$ but still less than 0.1 Å$^{-1}$) is consistent with any aggregates having sharp surfaces but does not otherwise specify shape[34]. The data from the SAXS plots provide supporting evidence that the laterally associated amyloid assemblies seen by AFM and TEM are not artefacts induced either by the dehydration (AFM), applying vacuum conditions (TEM) or the mica (AFM) or carbon (TEM) substrates used, but a genuine structure observed also in bulk.

Figure 3b shows the CD spectra of mature assemblies, of both peptides. Assemblies of ILLIIM display a quite simple spectrum indicating the dominance of β-sheets, with a minimum between 225 and 230 nm and a strong maximum at 205 nm[29]. The CD spectra of RNYIAQVD possess a well-defined minimum at 203 nm and a distinct shoulder at around 215 nm.

To further investigate the predicted secondary structure of both peptide assemblies we employed the secondary structure analysis software BeStSel (Supplementary Table 1)[35,36]. As expected from the classic shape of the spectra the predicted secondary structure of ILLIIM at 5 mg mL$^{-1}$ is exclusively made up of β-sheets (41.8%) and β-turns (58.2%). At these high concentrations, the composition of these β-sheets is shown to be exclusively left twisted, whilst at lower concentrations (1 mg mL$^{-1}$) a more complex mixture of right, left and non-twisted (relaxed) β-sheets are predicted. At both concentrations, the CD spectra of RNYIAQVD again suggest the secondary structure is dominated

by β-sheets; however, now they appear to be exclusively in the form of higher energy right-twisted β-sheets, similar to that observed in the highly strained structure of other amyloidogenic ultra-short peptides[30]. This additional strain introduced by β-sheets opposing the left-handed chirality seen in natural amino acids may explain the additional polymorphism and twisted microstructures seen in the AFM and TEM images of the RNYIAQVD assemblies (Fig. 2i and Supplementary Fig. 4a)[29]. The BeStSel fitting algorithm predicted the remainder of the RNYIAQVD secondary structure is composed of α-helical structure and further backbone conformations that could not be assigned (Supplementary Table 1). Part-helical CD spectra do not necessarily imply helical structure, especially considering that a single octapeptide cannot literally be 19% helix (two residues). Backbone conformation as reported by CD correlates through sheet structure to the assembled tertiary structure but no single level of organisation exclusively dictates any other, this is especially true in the case of coupling the twist of a peptide strand to the overall twist of the aggregate, which can relax to meet surface and shape-driven constraints through intersheet and interchain as well as intrachain degrees of freedom[29].

To further investigate the conformation of the amyloid assemblies formed we utilised the conformation-specific antibody A11 and the fluorescent probe thioflavin T (ThT). The former binds specifically to non-fibrillar amyloid oligomers and the latter is a commonly used molecular probe that becomes highly fluorescent when binding to amyloid assemblies[37]. As expected, when ILLIIM and RNYIAQVD assemblies were stained with ThT both demonstrated clearly visible fluorescent emission at 590 nm,

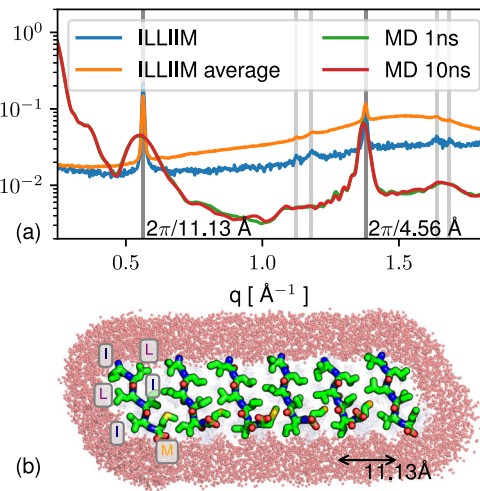

**Fig. 4 Molecular dynamics simulations of the ORF6 fragment, showing a proposed molecular unit cell that corresponds to the Bragg reflections from the WAXS.** Comparison of scattering peaks between atomistic model 2D amyloid sheet and solution SAXS/WAXS for ILLIIM. **a** The two major peaks (dark grey) agree well, and some agreement in second-order peaks (light grey) is also present. **b** The modelled 2D structures (parallel β hydrogen bonding axis goes into page) hydrophobic sidechains form a typical amyloid steric zipper. It is not proven that the solution structure is strictly 2D; however, the modelling does show that the SAXS/WAXS is consistent with an absence of full 3D symmetry in the solution. Assembly kinetics and cytotoxicity.

providing further evidence of their amyloid nature (Supplementary Fig. 9a, b). Conversely, A11 exhibited no positive binding; specifically, the level of fluorescence observed was similar to the background staining seen in the negative controls, as confirmed by similar fluorescent intensities for both assemblies (Supplementary Fig. 9e, f) and the negative controls (Supplementary Fig. 9d, h). Higher levels of A11 binding were seen for the positive control that consisted of phenylalanine assemblies known to form oligomeric species[38] (Supplementary Fig. 9g), with fluorescent intensities for these assemblies over 4 times greater than for ILLIIM or RNYIAQVD. Combined these data confirm the amyloid nature of the two ORF peptide fragments and suggest that non-fibrillar oligomeric amyloid species are absent.

The amyloid nature of the two assemblies is further confirmed by the wide-angle X-ray scattering (WAXS) spectra (Fig. 3d) of the peptide assemblies which possessed a number of strongly diffracting Bragg peaks. Both peptides have a clear peak at $1.38 \, \text{Å}^{-1}$ corresponding to a d-spacing of 4.6 Å which is indicative of an amyloid assembly composed of extended β-sheets[39]. It is worth noting that the apparent intrastrand spacing of ILLIIM assemblies is very slightly lower than the typically reported values (4.7–4.8 Å). This may be explained by the BeStCell analysis of the ILLIIM assemblies at 1 mg mL$^{-1}$, which suggests that the β-sheets in these assemblies are composed of a complex mixture of left-handed, strained right-handed and relaxed β-sheets (Supplementary Table 1); therefore, it is perhaps not surprising that the observed average intersheet spacing very slightly differs from that which is commonly reported. Furthermore, Lomont et al.[40] report that the observed intrastrand spacing from a range of amyloid crystal structures can vary by as much as 0.45 Å. ILLIIM also has a very strong Bragg reflection at 0.58 Å$^{-1}$ (11 Å) corresponding to a typical intersheet spacing given moderately bulky hydrophobic sidechains forming a steric zipper. RNYIAQVD has a number of well-defined Bragg peaks between 0.3 and 0.75 Å$^{-1}$ that are consistent with a mixture of first and second-order reflections

corresponding to an amyloid-like 3D symmetry. Typical reflections arising from the combinations of the longer two axes of the unit cell of short peptide amyloid crystals arise in the 0.3–0.75 Å$^{-1}$ region with a qualitatively similar appearance to the pattern from RNYIAQVD, although in this case the peaks could not be individually assigned.

Discovery of sub-Å resolution structures from solution WAXS is highly challenging; however, given the simple nature of the scattering from the ILLIIM system, it was possible to produce an atomistic model matching the positions of the observed peaks, although not their sharpness. Physically, peak sharpness increases with the ordering length scale, indicating that some structures in the solution were larger than could be managed in the simulation. The sheet-like shape factor and the presence of peaks at roughly $2\pi/4.6$ and $2\pi/11$ Å$^{-1}$ indicate assembly in solution dominated by the hydrogen bonding axis (with the typical parallel β sheet period of $\approx 4.7$ Å) and by a sidechain-sidechain hydrophobic zipper interface. A metastable candidate structure of size $6 \times 50 \times 1$ peptides was constructed following this geometry (see Methods) and found to reproduce the observed WAXS and to fully exclude water at the steric zipper (Fig. 4). The $q^{-2}$ dependence of ILLIIM scattering at low $q$ in solution (Fig. 3a) is consistent with a 2D sheet-like structure similar to that produced in the modelling. Initial assembly into sheets is also consistent with the eventual formation of multi-laminar structures as shown in the AFM (Fig. 2), as well as with the tendency of ILLIIM in particular to form lateral assemblies of needle microcrystals (Fig. 2g, k vs. e, i). Atomistic details of the interaction of the 2D sheet-like oligomer structure of Fig. 4 with neuronal cell membrane are difficult to predict and would be an interesting subject for further work. However, the juxtaposition of hydrophobic sidechains with polar termini in the ILLIIM fragment (or with titratable residues in the longer fragment E$_{13}$ILLIIMR, which unfortunately could not be synthesised) has a length of approximately 10 Å, comparable to the polar-hydrophobic-polar length scale of 40 Å for the two leaflets of the eukaryotic cell membrane, indicating a potential for planar aggregates of, in particular, four sheets in thickness (four peptides end-to-end, linked in the middle by salt bridges) to disrupt the cell membrane.

The ThT stain, which becomes highly fluorescent upon binding to β-sheet rich amyloid assemblies was used to assess the assembly kinetics of both ILLIIM and RNYIAQVD (Fig. 3c). Both peptides show rapid kinetics with significant assembly occurring almost instantaneously and reaching a plateau after 30–60 min. Longer amyloidogenic polypeptides typically show a distinct lag phase in their assembly kinetics; however, this was not observed in these sequences. This apparent lack of a lag phase in the assembly kinetics behaviour is typical of amyloidogenic short peptides, which have been previously seen to assemble very rapidly[29,41]. This is highly likely to be due to a lack of additional non-amyloidogenic amino acid sequences acting as a kinetic barrier to amyloid formation. The ThT signal for ILLIIM at 5 mg mL$^{-1}$ plateaus at about 300 a.u., this is slightly stronger than the maximum signal generated from mature fibrils of the somewhat homologous peptide ILQINS, which was around 250 a.u[29–31,42], suggesting that the amyloidogenicity of the two peptides is comparable. RNYIAQVD, whilst showing similar ThT values at low concentrations (1 mg mL$^{-1}$), generated a ThT signal nearly 3 times as large at 5 mg mL$^{-1}$ suggesting that the assembly of this peptide is highly concentration dependent. For reasons yet unknown, it seems that RNYIAQVD appears to reach a maximum ThT value and then begin to drop, this can be seen at both concentrations but is most obvious at the higher concentration. This could be due to a reversible self-assembly as seen in other functional amyloids[39,41,43,44], or to self-

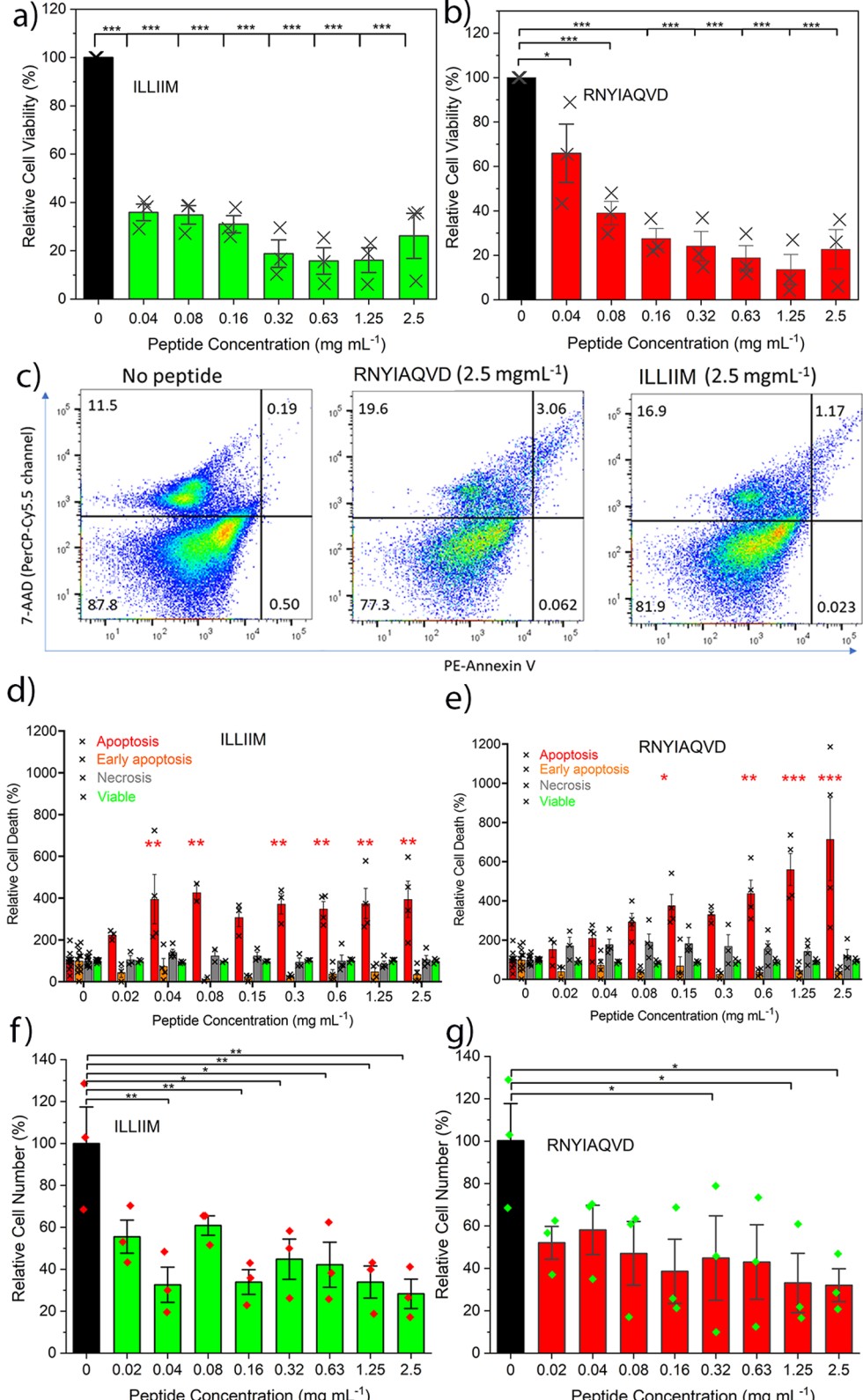

**Fig. 5 Cell metabolic and viability assays of ILLIIM and RNYIAQVD assemblies over a range of concentrations. a**, **b** MTT assays showing cell viability after 48 h incubation on cultured SH-SY5Y cells normalised to cells grown in the absence of peptide. **c** Representative flow cytometry plots of SH-SY5Y cells after 48 h culture with the amyloid assemblies, plots are split into four quadrants bottom left: viable, bottom right: early apoptosis, top left: necrotic, top right: late-stage apoptosis. **d**, **e** Quantification of cell death/viability pathways over a range of peptide assembly concentration normalised to cells grown in the absence of peptide (statistics are all relative to the no peptide control, and the result of at least three independent experiments). **f**, **g** Manual cell counts performed after 48 h incubation showing relative cell number compared to cells grown in the absence of peptide. For all panels, the error bars represent the standard error of mean of three biologically independent experiments ($n = 3$). Statistical analysis performed by one-way ANOVA with Tukey comparison. *$p < 0.1$, **$p < 0.01$, ***$p < 0.001$. Values of $p$ are given in Supplementary Table 2. FACS gating strategy given in the Supplementary information.

quenching of the amyloid bound aromatic ThT molecules, or simply to a reduction of exposed ThT-binding sites as larger aggregates with a lower surface area to volume ratio come to dominate the solution.

Cytotoxicity of both studied peptides also began to drop slightly (without statistical significance) at the highest concentrations tested (vide infra, Fig. 5), together with the drop in ThT response this supports the existence of a kinetically or thermodynamically available aggregate structure with reduced 'amyloid activity'. This is reminiscent of strongly amyloid correlated diseases such as AD, where the toxicity of amyloid can vary dramatically, with the relationship between the amount of amyloid deposited to the progress of the disease being idiosyncratic and highly non-linear[45]. Combined the CD spectroscopy (Fig. 3b), the ThT spectroscopy (Fig. 3c) and confocal microscopy (Supplementary Fig. 9), the presence of the Bragg peaks corresponding to the intra- and inter-β-sheet spacings (Fig. 3d) and the molecular modelling (Fig. 4) confirm beyond doubt the β-sheet rich, amyloid nature of these two fragments.

**ILLIIM and RNYIAQVD peptide assemblies are both highly toxic towards the neuroblastoma cell line SH-SY5Y.** Given the physical evidence and the discussions referred to in the introduction of various means by which SARS-CoV-2 and other viral infections could enhance their fitness (to the detriment of the host) by the production of amyloidogenic peptides, we hypothesised that the SARS-CoV-2 viral transcript fragments ILLIIM and RNYIAQVD are toxic to human neurons. This is in particular supported by the previously reported neuroinvasive capabilities of SARS-CoV-2[7,8], the noted similarities of the symptoms to a (hopefully transient form of) AD[5] and the previous detection of amyloid assemblies driven by other viruses[20]. To investigate this further we performed a number of cytotoxicity assays of the two peptide sequences against a human-derived neuroblastoma cell line (SH-SY5Y) often used as a model cell line for studying Parkinson's and other neurodegenerative diseases[46]. Using an MTT assay we found that cells grown in the presence of both peptide assemblies possessed much lower viability after 48 h incubation. Concentrations as low as 0.04 and 0.03 mM (for RNYIAQVD and ILLIIM, respectively) were seen to reduce the viability of cultured cells after 48 h to <50% (IC$_{50}$) compared to the cells cultured without the peptides (Fig. 5a, b). This toxicity in relation to concentration is similar to that reported for Aβ42[47] although expression levels and time-scales (sudden for COVID versus chronic for AD) are likely to be very different.

To gain further insight into the mechanism of cell death occurring in the peptide exposed cells, we performed a detailed flow cytometry analysis using the apoptotic stain Annexin V and the viability dye 7-AAD. Figure 5c shows representative flow cytometry plots; cells can be identified as viable (bottom left quadrant), viable but undergoing early apoptosis (bottom right), non-viable and necrotic (top left) or non-viable due to late-stage apoptosis (top right). The percentages of cells in these quadrants are roughly equal for all conditions tested except in the case of late-stage apoptosis where we see a large increase in the cells exposed to the peptide assemblies (a 6.25-fold increase for RNYIAQVD at 2.5 mg mL$^{-1}$). Quantification over a range of concentrations showed that on average cells exposed to both ILLIIM and RNYIAQVD had a 3–5-fold increase in late-stage apoptosis compared to SH-SY5Y cells cultured in the absence of peptide assemblies (Fig. 5d, e). No evidence of increasing necrosis was seen in any of the samples, suggesting that the amyloid assemblies are triggering programmed cell death via an apoptotic pathway. This triggering of late-stage apoptosis in the cells was more pronounced for ILLIIM than for RNYIAQVD, showing

statistically significant increases in apoptotic cells at concentrations as low as 0.04 mg mL$^{-1}$ for ILLIIM compared to 0.15 mg mL$^{-1}$ for RNYIAQVD. This increase in apoptosis down to low concentration provides convincing evidence, especially for ILLIIM, that the amyloid aggregates are responsible for this toxicity, as at these low concentrations we would expect very little un-assembled peptide to exist. The mechanisms of cell death in neurodegenerative diseases are complex and can vary between different diseases[48], and here we provide evidence that induction of apoptosis may be an important mechanism of neuronal death in COVID-19. Intriguingly, the conserved protein ORF6 from SARS-CoV-1 (not SARS-CoV-2) has previously been shown to induce apoptosis[49]. Furthermore, we performed a series of cell counting experiments and demonstrated that after 48 h incubation we saw statistically significant decreases in cell number for both peptides at concentrations as low as 0.04 mg mL$^{-1}$ for ILLIIM and 0.32 mg mL$^{-1}$ for RNYIAQVD. These results confirm that in addition to the cytotoxic nature of the peptide assemblies, they significantly reduce cell number especially in the case of ILLIIM. The significant increase in apoptosis and reduction in cell number seen for ILLIIM correlates with the work of Lee et al. who have previously shown that the ORF6 protein (that contains the ILLIIM sequence) is the most cytotoxic protein in the proteome of SARS-CoV-2[13]. Combined with our data, this suggests that this toxicity might be due to the amyloidogenic nature of this short protein.

Previous research has shown that the polymorphism, size distribution and the morphology of amyloid aggregates can have a large influence on their cytotoxicity. Marshall et al.[50] showed that a range of crystal-forming assemblies formed from short peptide sequences show surprisingly little toxicity to the same neuroblastoma cell line used in this study. Our TEM and AFM images (Fig. 2) confirm that the assemblies formed by the sequences identified from ORF6 and ORF10 look very similar to the assemblies in Marshall et al.[50] but the SARS-CoV-2-related peptides are significantly more toxic, suggesting a specific mechanism of toxicity for these assemblies. Xue et al.[51] showed that shorter amyloid assemblies from a range of different proteins/peptides have increased the ability to disrupt the bilayer of unilamellar vesicles and provide a greater cytotoxic effect on neuroblastoma cells. Mocanu et al.[52] showed a dose-dependent cytotoxic effect in epithelial cells for long-thin lysozyme amyloid fibrils, and a threshold dependent mechanism for the larger laterally associated fibrils. We see similar effects to both Xue et al. and Mocanu et al. suggesting that the observed toxicity of the assemblies may be related to their aspect ratio. We observed that ILLIIM assemblies are both more toxic, wider (Fig. 2h) and shorter than their RNYIAQVD counterparts (Fig. 2k), this is shown schematically in Fig. 6. Similarly to Mocanu et al.[52] we see that the long-thin RNYIAQVD fibrils show a clear dose-dependent increase in apoptosis (Fig. 5e), and the laterally associated ILLIIM fibrils show similarly high levels of apoptosis induction at all concentrations above a threshold of 0.04 mg mL$^{-1}$ (Fig. 5d).

There is a wealth of literature suggesting that in neurodegenerative diseases like Alzheimer's and Parkinson's amyloid oligomers are the main toxic culprits and mature amyloid fibrils are a more inert assembly end-point. This is seemingly at odds with our data; however, there have also been multiple studies that show mature assemblies can also display significant toxicity[37,53,54]. Alternatively, it may be the nature of the amyloids species seen here that differs from amyloids in neurodegenerative diseases; the amyloids seen here appear to be largely crystalline (especially in the case of ILLIIM). Previous work has shown that amyloid crystals are deeper in the free energy landscape compared to twisted protofilaments and amyloid ribbons[30,55], representing a global energy minima. AFM and TEM data have

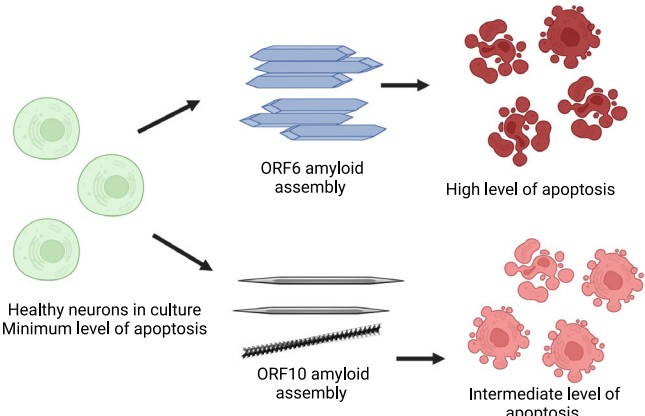

**Fig. 6 Amyloid assemblies formed from ORF6 and ORF10 fragments cause cell death to neurons via an apoptotic pathway.** A greater proportion of cells undergo late-stage apoptosis when incubated with ground energy ORF6 amyloid crystals compared to the higher free energy twisted amyloid assemblies sometimes formed from ORF10 fragments, suggesting a direct correlation between amyloid polymorphism and cytotoxicity.

shown that these stable amyloid crystals are the dominant polymorph for ILLIIM (Fig. 2c, g, k) and that RNYIAQVD shows examples of higher energy (partially) twisted fibrils (Fig. 2i and Supplementary Figs. 4a and 6). Therefore, we hypothesise that the low energy ILLIIM crystalline assemblies are more slowly metabolised and cells are exposed for longer timeframes to the cytotoxic effect compared to RNYIAQVD assemblies. To date, there have been few investigations into the toxicity of amyloid crystals compared to other more commonly reported amyloid species. For the reasons above, the toxic nature of these amyloid assemblies warrants further investigations into the potential presence of amyloid aggregates from SARS-CoV-2 in the CNS of COVID-19 patients, and the potential role of amyloids in the neurological symptoms observed.

In conclusion, using a bioinformatics approach we identified two strongly amyloidogenic sub-sequences from the ORF6 and ORF10 sections of the SARS-COV-2 proteome. Nanoscale imaging, X-ray scattering, molecular modelling, spectroscopy and kinetic assays revealed that these self-assembled structures are amyloid in nature, and screening against neuronal cells revealed that they are highly toxic (approximately as toxic as the toxic amyloid assemblies in AD) to a cell line frequently used as a neurodegenerative diseases model. The neuroinvasive nature of SARS-COV-2 has been established previously[7,8]; therefore, it is entirely plausible that amyloid assemblies either from these ORF proteins or other viral proteins could be present in the CNS of COVID-19 patients. The cytotoxicity and protease-resistant structure of these assemblies may result in their persistent presence in the CNS of patients post-infection that could partially explain the lasting neurological symptoms of COVID-19, especially those that are novel in relation to other post-viral syndromes such as that following the original SARS-CoV-1. The outlook in relation to triggering of progressive neurodegenerative disease remains uncertain. Given the typically slow progress of neurodegenerative disease if such a phenomenon exists, it will most probably take some time to become evident epidemiologically.

## Methods

**Amyloid prediction algorithms.** The online amyloid prediction algorithms TANGO and ZIPPER were used to predict peptide sequences with a tendency to form β-rich amyloid assemblies. TANGO is an algorithm that predicts aggregation nucleating regions in unfolded polypeptide chains[56]. It works on the assumption

that the aggregating regions are buried in the hydrophobic core of the natively folded protein. ZIPPER is an algorithm that predicts hexapeptides within larger polypeptide sequences that have a strong energetic drive to form the two complementary β-sheets (known as a steric zipper) that give rise to the spine of an amyloid fibril[57]. Both methods are physically motivated but rely on statistically determined potentials.

**Self-assembly of peptides.** NH2-ILLIIM-CO2H and Ac-RNYIAQVD-NH2 (>95% pure) were purchased from GL Biochem Ltd (Shanghai, China). Ideally, it would have been preferred to have both peptides capped (N-terminus: Acetyl and C-terminus: Amide), as they would better represent small fragments of a larger peptide sequence. Due to the fact ILLIIM contains no charged sidechains, synthesising capped sequences to high purity would have been very challenging, therefore only the RNYIAQVD sequence remained capped and the ILLIIM sequence had regular carboxyl and amino termini. To ensure that all peptide seeds were fully dissolved before self-assembly was initiated the peptides were solubilised in warmed PBS (90 °C) at either 1 or 5 mg mL$^{-1}$ The warmed peptide solutions were vortexed vigorously and held at 90 °C for 3 h to ensure maximum dissolution. After the second round of vortexing, the peptide suspensions were cooled slowly. This protocol has been previously used to maximise a homogenous starting population of monomeric peptide[32]. Alternatively, self-assembly was carried out at a constant temperature of 37 °C without pre-solubilising the peptides in hot PBS.

**Atomic force microscopy (AFM).** AFM imaging was performed on a Bruker Multimode 8 AFM and a Nanoscope V controller. Tapping mode imaging was used throughout, with antimony (n)-doped silicon cantilevers having approximate resonant frequencies of 525 or 150 kHz and spring constants of either 200 or 5 Nm$^{-1}$ (RTESPA-525, Bruker or RTESPA-150, Bruker). No significant differences were observed between cantilevers. 50 μL aliquots of the peptide (either at 1 or 5 mg mL$^{-1}$) were drop cast onto freshly cleaved muscovite mica disks (10 mm diameters) and incubated for 20 min before gently rinsing in MQ water and drying under a nitrogen stream. All images were flattened using the first order flattening algorithm in the nanoscope analysis software and no other image processing occurred. Statistical analysis of the AFM images was performed using the open-source software FiberApp[33] from datasets of no less than 900 fibres.

**Transmission electron microscopy (TEM).** Copper TEM grids with a formvar-carbon support film (GSCU300CC-50, ProSciTech, Qld, Australia) were glow discharged for 60 s in an Emitech k950x with k350 attachment. Then, 5 μL drops of sample suspension were pipetted onto each grid, allowed to adsorb for at least 30 s and blotted with filter paper. Two drops of 2% uranyl acetate were used to negatively stain the particles with excess negative stain removed by blotting with filter paper after 10 s each. Grids were then allowed to dry before imaging. Grids were imaged using a Joel JEM-2100 (JEOL (Australasia) Pty Ltd) transmission electron microscope equipped with a Gatan Orius SC 200 CCD camera (Scitek Australia).

**Small- and wide-angle X-ray scattering (SAXS/WAXS).** SAXS/WAXS experiments were performed at room temperature on the SAXS/WAXS beamline at the Australian Synchrotron. Peptide assemblies in PBS prepared at both 1 and 5 mg mL$^{-1}$ were loaded into a 96-well plate held on a robotically controlled $x$-$y$ stage and transferred to the beamline via a quartz capillary connected to a syringe pump. Data from the 5 mg mL$^{-1}$ assemblies were discarded due to sedimentation of the assemblies preventing reliable sample transfer into the capillaries. The experiments used a beam wavelength of $\lambda = 1.03320$ Å$^{-1}$ (12.0 keV) with dimensions of 300 μm × 200 μm and a typical flux of $1.2 \times 10^{13}$ photons per second. 2D diffraction images were collected on a Pilatus 1M detector. SAXS experiments were performed at $q$ ranges between 0.002 and 0.25 Å$^{-1}$ and WAXS experiments were performed at a $q$ range between 0.1 and 2 Å$^{-1}$. These overlapping spectra provide a total $q$ range of 0.002–2.2 Å$^{-1}$. Spectra were recorded under flow (0.15 mL min$^{-1}$) to prevent X-ray damage from the beam. Multiples of approximately 15 spectra were recorded for each time point (exposure time = 1 s) and averaged spectra are shown after background subtraction against PBS in the same capillary.

**Circular dichroism spectroscopy.** CD spectroscopy was performed using an AVIV 410-SF CD spectrometer. Spectra were collected between 190 and 260 nm in PBS using 1 mm quartz cuvettes with a step size of 0.5 nm and 2 s averaging time. Data were analysed using the BeStSel (Beta Structure Selection) method of secondary structure determination[35].

**Atomistic modelling.** Atomistic models were constructed using the Nucleic Acid Builder[58]. Simulations were run in explicit water (TIP3P[59]) using the ff15ipq forcefield[60] and the pmemd time integrator[61]. In order to hold the unit cell geometry to values consistent with the observed scattering, the alpha carbon of the central residue of each chain was subjected to a restraining force with spring constant 2 kcal mol$^{-1}$ Å$^{-2}$. Periodic boundaries were applied to the system such that it formed a truncated octahedron, which was relaxed during equilibration to a volume of 10,310 nm$^3$, giving a density of 0.973 reference with 323,535 water molecules. The system state after 10 ns of equilibration was stripped of water

molecules more than 10 Å from any non-hydrogen solute atom, and passed to CRYSOL3 for calculation of orientationally averaged scattering profile given the example state (including the ordered waters from the explicit solvent shell, and also including an approximate treatment of ordered water beyond this shell)[42,62].

**Thioflavin T amyloid kinetic assays.** Peptide assemblies were made up to concentrations of 1 or 5 mg mL$^{-1}$ suspensions containing 25 μM ThT in PBS. The first fluorescence measurement ($t = 0$) was recorded immediately after sample preparation. All the samples were then stored at room temperature and fluorescence intensity was recorded at different time points. Measurements were performed in triplicate using a ClarioStar fluorimeter equipped with a 96-well plate reader (excitation wavelength: 440 nm, emission wavelength: 482 nm).

**Cell line and cultures.** Human-derived neuroblastoma cells (SH-SY5Y, ATCC Product Number: CRL-2266) were cultured in DMEM-F12 (Invitrogen) medium supplemented with 10% (v/v) foetal calf serum (FCS), 100 UmL$^{-1}$ penicillin and 100 μgmL$^{-1}$ streptomycin (Invitrogen, Carlsbad, CA). Cells were cultured at 37 °C in a humidified atmosphere containing 5% $CO_2$.

**Immunofluorescent and thioflavin T microscopy.** ThT staining was performed by incubating the amyloid assemblies with a 25 μM solution of ThT in PBS for 15 min, in an 8-well Labtek II chamber (Nunc). For the antibody stain the same assemblies were incubated in a 1:200 dilution of the A11 polyclonal antibody raised in rabbit (Invitrogen, REF: AHB0052, LOT:VF299837) in 2% BSA in PBS for 1 h. Following this the wells were carefully washed in PBS and a solution of the 2° antibody (Goat-Anti Rabbit IgG-Alexa Fluor 647, Product A21244, Lot #: 1871168, Molecular Probes) at 1:1000 dilution in PBS was incubated with the peptides for 1 h. Finally the assemblies were once again washed in PBS before being imaged via laser scanning confocal microscopy using a FV3000 microscope (Olympus) and 60× objective lens (1.35 NA Oil Plan Apochromat) using the following settings: ThT channel λex = 450 nm, λem = 490 nm, Alexa Fluora 647 channel λex = 650 nm, λem = 665 nm. The same imaging settings were used for all samples and the negative controls (peptide assemblies + 2° antibody only) were used to determine the level of background fluorescence. For positive controls, amyloid assemblies of phenylalanine were used under concentrations known to readily form oligomeric amyloid assemblies[38], which were shown here to bind strongly to the A11 antibody (Supplementary Fig. 9e, f).

**Cell viability assay.** Cells were seeded into 96-well plates at $1 \times 10^5$ cells per mL and incubated for 24 h to ensure good attachment to the surface. A stock solution of peptide assemblies (10 mg mL$^{-1}$) was serially diluted into DMEM-F12 media 2.5–0.02 mg mL$^{-1}$ or 3.3–0.027 mM for ILLIIM and 2.45–0.02 mM for RNYIAQVD) and seeded onto the SH-SY5Y cells and incubated for 48 h, cell viability was determined using 3-(4,5-dimethylthiazol-2-yl)2,5-diphenyltetrazolium bromide (MTT) (Sigma-Aldrich) as described previously[63]. Equivalent MTT assays were performed on cells cultured in the same ratios of PBS to media, but in the absence of peptide, assemblies to confirm that the culture conditions were non-toxic (Supplementary Fig. 10). Absorbance readings of untreated control wells in 100% cell culture media were designated as 100% cell viability. Statistical analysis was performed by one-way ANOVA tests with Tukey comparison in the software GraphPad (Prism) ***$p < 0.001$. Flow cytometry assays to determine cell viability were performed in a similar manner to the MTT assays. Briefly, to determine the effect of the peptides on cellular viability SH-SY5Y cells were cultured in the presence of the peptides for 48 h, harvested and stained with the apoptosis stain Annexin V for 10 min on ice (Cat No. 550474, BD Biosciences, 5 μL in 100 μL of 2% FCS in PBS). Samples were diluted with 300 μL of 2% FCS in PBS and stained with the viability dye 7-AAD (559925 BD Biosciences, 5 μL per sample) and analysed using flow cytometry (FACS Aria III; BD Biosciences). Cell counts were performed manually using a hemocytometer, with tryphan blue to differentiate non-viable cells.

**Reporting summary.** Further information on research design is available in the Nature Research Reporting Summary linked to this article.

## Data availability
The authors declare that all the data supporting the findings of this study are provided in the Supplementary Information and Source Data file.

## Code availability
All code used in this study is either free (NAB 1.3, pymol 2, TANGO 2.2 and ZIPPER) or commercially available (pmemd 19, CRYSOL3 3.0.3).

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

## Acknowledgements

N.P.R. would like to acknowledge The La Trobe Institute of Molecular Sciences (LIMS) for the receipt of a Nicholas Hoogenraad fellowship, and the CASS foundation for partially funding this work through a philanthropic grant (#10053, 'Determining the role of protein aggregation in COVID-19'). N.P.R. would also like to acknowledge that Fig. 6 was created using Biorender.com. The authors thank Dr Susi Seibt for assistance on the SAXS/WAXS beamline at the Australian Synchrotron. This research was undertaken, in part, on the SAXS/WAXS beamline at the Australian Synchrotron, part of ANSTO. Molecular dynamics calculations made use of the HPC service of the University of Luxembourg[64]. The project was part-funded by grant C20/MS/14588607 of the Fonds Nationale de la Recherche, Luxembourg.

## Author contributions

M.C. performed the biological and synchrotron experiments and co-wrote the manuscript. S.I. performed the spectroscopic and bioinformatic analysis. G.K.B. performed biological experiments and edited the manuscript. J.E. performed cell culture for the paper. J.R. performed the electron microscopy imaging and editing of the paper. J.Z. contributed to the atomic force microscopy imaging and editing of the paper. R.M. edited the paper and co-conceptualised the project. M.D.H. edited the paper, provided cell culture reagents and lab space. K.H. performed molecular modelling and edited the paper. J.T.B. co-conceptualised, co-wrote the manuscript and performed molecular simulations. N.P.R. co-conceptualised, co-wrote and performed atomic force microscopy and synchrotron experiments.

## Competing interests

The authors declare no competing interests.
