## [Peer Review File · Nature Communications]

REVIEWER COMMENTS

Reviewer #1 (Remarks to the Author):

The research article submitted by Charnley et al., titled “Neurotoxic Amyloidogenic Peptides in the Proteome of SARS-COV2: Potential Implications for Neurological Symptoms in COVID-19,” outlines the presence of aggregation-prone regions in SARS-CoV-2 capsid and other protein and their relevance in neuropathology behavior during COVID-19 infection. The investigators hypothesized that following COVID-19 infection, many viral particles are formed, which are subsequently degraded by the immune system. During proteolytic degradation of viral proteins, the amyloidogenic sequences might release in the biological milieu and undergo intermolecular interactions to form amyloid-like structures, which subsequently might be involved in neurodegenerative events similar to the involvement of Amyloid beta peptides in Alzheimer’s disease. In this study, the investigators have used two aggregation-prone peptides (i.e., NH₂-ILLIIM-CO₂H and Ac-RNYIAQVD-NH₂) based upon in silico prediction and tested their amyloidogenic potential under in vitro conditions. The data partially support that these peptides form amyloid-like structures and possess neurotoxicity. The hypothesis appears to be quite relevant under the current scenario, and the study might be of some importance. However, the data presented here are not adequate to justify the claims. For examples;

1. The manuscript does not have a single in vivo data that confirms the formation of amyloid-like structures during COVID-19 infection. Or such virus-derived aggregation-prone peptides exist in the COVID-19 patients.
2. It is unclear how the neutralized viral particles are degraded and how the aggregation-prone peptides are released. The respiratory airways and alveoli are considered as the primary site of viral abundance. Then how are such peptides formed in the lungs and transferred to the brain tissue of the patients to exert neuropathological consequences?
3. Many of the data lack appropriate controls. For example, in Figure 3B, a spectrum corresponding to the freshly dissolved un-aggregated peptides is missing. The cell viability assay does not have the data corresponding to freshly dissolved un-aggregated peptides.
4. The characterizations of the amyloid-like aggregates are inadequate and inconsistent. The CD data of one of the peptides suggested the presence of beta-sheets, but the other showed a random coil. An amyloid aggregate of the random coil is unrealistic.
5. Conventionally, out of three significant conformers, oligomers, protofibrils, and fibrils, the amyloid-induced neurotoxicity is primarily demonstrated by the oligomers and not due to fibrils. The authors do not advocate it.
6. The ThT kinetic data presented in Figure 3C does not represent a typical aggregation pattern of a specific amyloidogenic peptide. It seems that immediately after being suspended in the buffer, the peptide immediately formed amyloid, which is less likely. Such patterns are commonly observed during amorphous aggregation. Immediately after suspension, the peptide forms a milky white precipitate and

stays for some time. The formation of amyloids generally occurs through ordered polymerization of beta-sheet depicting the lag, log, and saturation phases on the sigmoidal aggregation kinetics.

7. Finally, the manuscript lacks mechanistic dimensions of their observation and interpretation.

On the other hand, the manuscript is written very well in crystal clear manner. The manuscript may be accepted after judicious addressal of the above comments.

Reviewer #2 (Remarks to the Author):

In the present study, authors predicted the peptide fragments (ILLIIM and RNYIAQVD) in the SARS-CoV-2 proteome to be highly amyloidogenic using a bioinformatic scan. The study reported two peptides, ILLIIM and RNYIAQVD, from the open reading frame 6 (ORF6) and open reading frame 10 (ORF10) proteins, respectively, which rapidly self-assemble into amyloid aggregates and the amyloid assemblies displayed toxicity to a neuronal cell line. The authors suggested that the cytotoxic amyloid aggregates of SARS-CoV-2 proteins are responsible for the neurological symptoms observed in COVID-19 patients. The authors need to address the following queries:

1. On page 6, authors mentioned “The CD spectra of RNYIAQVD are less obvious and seem to resemble the typical spectra of a random polypeptide coil, except in that there is a well-defined minimum at 203 nm (a typical random coil is < 200 nm) and additional shoulder appears at around 215 nm.” However, the secondary structure analysis using BeStSel highlighted that RNYIAQVD sampled a complex mixture of secondary structures, which are dominated by β -sheets and β -turns (totalling 42 %). The authors need to comment on why random coil conformation of RNYIAQVD was observed in the CD spectra as compared to the dominance of the β -sheets and β -turns conformations in the secondary structure analysis using BeStSel?

2. The authors mentioned on page 6 that the CD spectra of mature assemblies of ILLIIM display a quite simple spectrum indicating dominance of β -sheets, whereas the CD spectra of RNYIAQVD are less obvious and seem to resemble the typical spectra of a random polypeptide coil. ILLIIM having higher sampling of the β -sheet conformation should be more aggregation-prone as compared to RNYIAQVD having higher propensity for the random coil conformation. However, ThT assay highlighted higher aggregation tendency of RNYIAQVD as compared to ILLIIM (the ThT signal for ILLIIM at 5 mg mL⁻¹ plateaus at about 300 a.u., whereas RNYIAQVD generated a ThT signal nearly 3 times of the ThT signal observed in ILLIIM at the same concentration, 5 mg mL⁻¹). The authors need to comment on the difference in the observed aggregation tendencies of the two peptides in the CD and ThT fluorescence spectra for the better clarity of the readers.

3. The authors mentioned on page 4, “Simulations were run in explicit water (TIP3P37) using the ff15ipq forcefield38 and the pmemd time integrator39.” However, authors have not discussed the results of the

simulations in the manuscript draft. Moreover, the details about the simulation length, box size, number of water molecules etc. should be included in the manuscript text.

4. The authors need to provide the full form of BeStSel in the manuscript text.

Reviewer #3 (Remarks to the Author):

To authors:

In this study, the authors hypothesized that the amyloidogenic potential of the viral protein is responsible for the neurological symptoms caused by SARS-CoV-2. They searched for candidate amino acid sequences in the viral protein for amyloidogenicity, showed that the protein actually aggregated and demonstrated that the aggregates are amyloidogenic and cytotoxic at the level of cultured cells.

This study extensively demonstrates that this novel candidate sequence is amyloidogenic by various analytical techniques. However, it has been known that partial peptides of proteins can be amyloidogenic.

In addition to lysozyme, which the authors describe in the main text, P8 (p53(250–257)) in the tumor suppressor p53 has been reported to be amyloidogenic (<https://doi.org/10.1021/bi500825d>).

Nevertheless, SARS-CoV-2 is now sweeping the world, and findings on its relevance to existing concepts other than viral infections are highly impactful.

In this context, in my opinion, to attract a wide audience of Nature Communications, this study should report the following points to show the relevance of this study to a possible phenomenon in vivo.

Does the sequences selected for this study indicate any of the following?

1. is there any evidence that fragmented peptides of ORF6 and ORF10 proteins exist in vivo?
2. do full-length or in vivo processed ORF6 and ORF10 proteins form amyloid?
3. do ILLIIM and RNYIAQVD cause amyloid aggregation of ORF6 and ORF10 proteins in their native forms (full-length or processed length) in vivo?

Minor comments:

1. this study should explain the rationale for not selecting D30YIINLIKLN.
2. does it form amyloid fibrils at physiological temperatures? in the M&M's description, the temperature is gradually lowered from 80°C, and it appears that fibrils are not formed unless the temperature is high.
3. ILLIIM and RNYIAQVD can coexist in vivo. Do these peptides show cross-seeding?
4. for AFM images, please describe the scanning range and the number of scanning lines in the y-direction, the number of pixels in the x-direction, the color scale and the corresponding z range in the Figure caption so that the reader can understand that spatial resolution.
5. for the histogram in Figure 2, please provide the equation for the fitting curves and the fitting parameters in the figure legend.
6. for the AFM analysis, the main text describes the thickness step of ILLIIM as 12 nm, but in Figure S3 it looks clearly smaller than 12 nm. The appropriate explanation should be added.
7. RNYIAQVD has twisted fibers in the upper left corner of Figure S4 and Figure 2i. This paper also explained that the distribution of layers is not a single step. RNYIAQVD can have structural polymorphisms, so please explain this point as well.

Reviewer #4 (Remarks to the Author):

This paper presents a study of two sub-sequences of the SARS-COV-2 proteome which the authors believe form amyloid fibres. They conclude that if these form in the brain, they might be significant for long Covid symptoms.

I was asked specifically to review the SAXS/WAXD data presented in this paper. The two peptide sequences explored show the formation of needle like crystals which aggregate to form large assemblies and are adequately described by AFM and TEM. I find that the SAXS results do not contribute quantitatively to the findings. In addition, the SAXS experiments are conducted over a very limited q-range which is very moderate even for lab-based SAXS measurements. The q-range is from 0.02 to 0.25 Å⁻¹, which means that any quantitative information for the size of aggregates is impossible.

For the two peptide sequences, a qualitative difference in the SAXS curve is observed and yet these differences are merely reported and not discussed further. The RNYIAQVD aggregates show a q⁻⁴ dependence whereas the ILLIIM shows a q⁻² dependence. This difference is likely to arise from the ILLIIM aggregates having a broader lateral dimension as described. The q⁻⁴ dependence could arise from a sharp interface between a scattering entity and its background, however, there is no discussion that this may be different for the two peptides. Thus I conclude that the SAXS data as discussed do not contribute to the paper.

For the WAXS data, these were measured on the aggregates in solution, probably simultaneously as the SAXS data was collected. Wide-angle diffraction data from isolated needle crystals would of course provide much richer information. The data show two diffraction peaks for ILLIIM which the authors identified as likely from a beta-sheet structure. The inter-sheet spacing was calculated to be 4.8 Å but typically this is often measured at 4.7 Å. No reason for this discrepancy is given. The cross beta-sheet peak at approx. 11 Å is sharp in comparison to the simulation. No explanation for this finding is given. Interpreting this as a beta-sheet structure is only possible with confidence when taking into account the CD data as well.

For RNYIAQVD a more complex WAXS pattern is shown without any indexing of the diffraction peaks. The authors comment that “RNYIAQVD has a number of well-defined Bragg peaks in between 0.3-0.8 Å⁻¹ which are consistent with a mixture of first and second order reflections corresponding to an amyloid-like 3D symmetry.” In the future, they should show how they arrive at this conclusion through indexing and referencing appropriately. A 3D molecular model may help in this assertion.

In conclusion I find that the SAXS/WAXS data do not contribute to the scientific merit of this paper. Differences in the amyloid structures for the two peptide sequences are shown from AFM, TEM and CD data alone, However, the paper fails to explore the significance of these differences upon the cytotoxicity. Indeed, the structural information provided by these physical characterization methods only serves to point to a possible amyloid structure at the peptide concentration used in these methods. This is a key weakness in this paper to not interpret their findings further. I feel it is not acceptable to present, in a journal of this standing, results without further exploring their impact on the conclusions drawn. Therefore, for the physical characterisation of amyloid fibrils, I would not recommend publication.

Reviewer 1:

"The manuscript does not have a single in vivo data that confirms the formation of amyloid-like structures during COVID-19 infection. Or such virus-derived aggregation-prone peptides exist in the COVID-19 patients."

Whilst this statement is true, we argue that performing these experiments would be incredibly difficult and time consuming and therefore beyond the scope of this paper. This is largely as, to the best of the authors knowledge, there are no reliable animal models of COVID-19 infection therefore these experiments would have to be done on human participants. Further, amyloid deposits are almost impossible to detect in live human brains therefore post-mortem virally infected brain tissue would be required. This would require extensive ethics approval which can take up to 12 months to process, further none of our or our collaborators labs possess the necessary biosafety levels (PC3) to handle tissues and materials infected with SARS-CoV-2. We suggest that it would be better to share this data with the scientific community now, so that researchers with more expertise in working with virally infected human tissue can do these important experiments.

"It is unclear how the neutralized viral particles are degraded and how the aggregation-prone peptides are released. The respiratory airways and alveoli are considered as the primary site of viral abundance. Then how are such peptides formed in the lungs and transferred to the brain tissue of the patients to exert neuropathological consequences?"

We thank the reviewer for their critical reading of the manuscript, however we do not mean to propose that the amyloid assemblies are formed due to degradation of a viral particle. We suspect that the ORF6 and ORF10 viral transcripts, which are not part of the viral particle, but which are expressed from the viral genome in order to carry out some as yet undetermined function, form neurotoxic amyloids. Whilst it is true that the respiratory system is the primary site of viral abundance, by now it is well reported that SAR-CoV-2 infection is systemic and there is evidence that SARS-CoV-2 is neuroinvasive, with either the full virion or fragments of virus having been found in the CNS of infected patients. The presence of viral fragments in the bloodstream and CNS eliminates a need for a mechanism of formation in the lungs and transference to the brain (see page 2, lines 2-30).

"Many of the data lack appropriate controls. For example, in Figure 3B, a spectrum corresponding to the freshly dissolved un-aggregated peptides is missing. The cell viability assay does not have the data corresponding to freshly dissolved un-aggregated peptides."

Due to the very rapid aggregation kinetics of these peptides it is impossible to provide these controls, indeed in the ThT assays we see aggregation occurring just seconds after hydration. These rapid aggregation kinetics occur even if the peptides are directly added to water or PBS at room temperature. Such rapid aggregation kinetics are very common in crystalline amyloid forming systems (e.g. see Reynolds *et. al.*, Nature Comms, 2017). The fast ThT kinetics clearly show that an aggregated state is favorable for these assemblies. Therefore, in order to try to provide evidence that it is the amyloid resulting in this toxicity, we have repeated the Annexin/PE viability experiments over a larger range of concentrations. We now see a clear increase in apoptosis occurring even at the lowest concentrations, where we would expect there to be exclusively aggregated peptide (due to the combination of fast kinetics and low initial peptide concentrations). We believe this additional biochemical data provides convincing evidence that the aggregated form of the peptide is the cause of apoptosis in these cells. See Page 8, line 3-17 and Page 8, lines 46-50.

“The characterizations of the amyloid-like aggregates are inadequate and inconsistent. The CD data of one of the peptides suggested the presence of beta-sheets, but the other showed a random coil. An amyloid aggregate of the random coil is unrealistic.”

We apologize for the unintentional confusion caused with our discussion of a random coil structure from the CD data. We agree that amyloid aggregate by definition cannot be composed of peptides with a random coil. Whilst the writing in this part of the manuscript was not as clear as it could have been we meant to suggest that the CD spectra qualitatively looked similar to that of a random coil, but with subtle differences (such as no positive ellipticity, and no minima below 200 nm). Further interpretation and secondary structure analysis revealed that the spectra is explained by a number of different polymorphic beta sheets all contributing to a complex CD spectra (see Figure S9). We have now significantly reworded our discussion of the CD spectra emphasizing that both spectra arise from beta sheet rich amyloid assemblies. We have made additional edits to the manuscript to clarify this point (see page 6, lines 37-page 7, line 8)

“Conventionally, out of three significant conformers, oligomers, protofibrils, and fibrils, the amyloid-induced neurotoxicity is primarily demonstrated by the oligomers and not due to fibrils.”

We thank the reviewer for their comments, indeed the modern consensus in neurodegenerative diseases like Alzheimer’s and Parkinson’s is that oligomeric species are more toxic than mature amyloid fibrils. However, there are a number of robust studies that clearly show mature amyloid species can also display significant toxicity (see refs 60-62 for examples). Furthermore, the dominant species in our study seems to be amyloid nanocrystal/microcrystal (as previously detected in Lara et al, JACS, 2014, Reynolds et al, Nature Comms, 2017 etc). The toxicity of these amyloid crystals is not as well studied as the more well-known conformers. We have added additional sentences to the results and discussion to comment on these differences to traditional amyloid fibril toxicity studies, see page 9, lines 27-38.

“The ThT kinetic data presented in Figure 3C does not represent a typical aggregation pattern of a specific amyloidogenic peptide. It seems that immediately after being suspended in the buffer, the peptide immediately formed amyloid, which is less likely. Such patterns are commonly observed during amorphous aggregation. Immediately after suspension, the peptide forms a milky white precipitate and stays for some time. The formation of amyloids generally occurs through ordered polymerization of beta-sheet depicting the lag, log, and saturation phases on the sigmoidal aggregation kinetics.”

The reviewer here is describing the amyloid kinetics of a well-studied amyloidogenic polypeptide such as alpha-synuclein or IAPP. Rapid amyloid kinetics such as what we see here are common in amyloid forming ultra-short peptides and have been frequently reported previously (Lara et al, JACS, 2014 and Reynolds et al, Nature Comms, 2017). In fact, the lack of a lag phase in these systems is expected as there is no non-aggregating polypeptide sequence to slow down the assembly kinetics. Once again the fact that we are observing amyloid crystals which do not follow the same well-established behavior as amyloid forming proteins (or large peptides) is important. As this is an important distinction well raised by the reviewer we have tried to clarify the section of the results and discussion discussing the lack of lag phase typically observed with amyloidogenic ultra-short peptides, hopefully making it clear that these assemblies do not display traditional amyloid formation kinetics/cytotoxic behaviors of longer peptides. See page 8 lines 3-16. We thank the reviewer for inspiring us to add this important clarification.

“Finally, the manuscript lacks mechanistic dimensions of their observation and interpretation”

We have added significant mechanistic discussions and references to previous works which link the enhanced toxicity of the ILLIIM fragments with their observed shorter morphologies. Further we now observe concentration dependent increases in apoptosis induced by the longer thinner RNYIAQVD assemblies, and equivalent levels of toxicity above a threshold concentration for the lower aspect ratio ILLIIM. These findings have previously been reported for amyloid assemblies from α -synuclein and lysozyme of varying lengths and aspect ratio’s, suggesting a similar size dependent mechanism of toxicity occurring in our assemblies. As it is known that amyloid crystals possess a lower free energy than fibrils or other conformers (see Reynolds et al, Nature Comms, 2017). We also propose that the additional stability of the ILLIIM crystal polymorph, maybe harder to metabolize, by cells, than the higher energy RNYIAQVD polymorphs (consisting of a mixture of crystals and twisted fibrillar structures),

leading to increased cytotoxicity. We suggest these findings may be important as it could influence the design of therapeutic strategies that may look to modulate the shape of formed amyloid assemblies to reduce a neurotoxic effect occurring during infection. See page 9, lines 3-38 for additional content in the revised manuscript.

Reviewer 2

“On page 6, authors mentioned “The CD spectra of RNY!AQVD are less obvious and seem to resemble the typical spectra of a random polypeptide coil, except in that there is a well-defined minimum at 203 nm (a typical random coil is < 200 nm) and additional shoulder appears at around 215 nm.” However, the secondary structure analysis using BeStSel highlighted that RNY!AQVD sampled a complex mixture of secondary structures, which are dominated by /3-sheets and /3-turns (totalling 42 %). The authors need to comment on why random coil conformation of RNY!AQVD was observed in the CD spectra as compared to the dominance of the /3-sheets and /3-turns conformations in the secondary structure analysis using BeStSel?”

Once again we apologize for comparing the spectra of the RNYIAQVD assemblies to that of a random coil, the language used by us was unclear and potentially misleading, as described above we were attempting to highlight the fact that the CD spectra looks similar to that of a random coil, but actually as shown by the secondary structure analysis algorithms was in-fact predominantly composed of **1**-sheets as would be expected from an amyloid system. We have changed the language of this discussion to make it clear we never intended to suggest the CD data was inferring a random coil structure. (see page 6, lines 44 page 7 line 8).

“The authors mentioned on page 6 that the CD spectra of mature assemblies of !LL!!M display a quite simple spectrum indicating dominance of /3-sheets, whereas the CD spectra of RNY!AQVD are less obvious and seem to resemble the typical spectra of a random polypeptide coil. !LL!!M having higher sampling of the /3-sheet conformation should be more aggregation-prone as compared to RNY!AQVD having higher propensity for the random coil conformation. However, ThT assay highlighted higher aggregation tendency of RNY!AQVD as compared to !LL!!M (the ThT signal for !LL!!M at 5 mg mL⁻¹ plateaus at about 300 a.u., whereas RNY!AQVD generated a ThT signal nearly 3 times of the ThT signal observed in !LL!!M at the same concentration, 5 mg mL⁻¹). The authors need to comment on the difference in the observed aggregation tendencies of the two peptides in the CD and ThT fluorescence spectra for the better clarity of the readers.”

We have significantly altered the discussion of the CD data, added more secondary structure analysis (figure S9) and included the fitting parameters used for the secondary structure analysis to address this comment. In summary, our new comprehensive analysis shows both peptide assemblies have approximately the same **1** sheet content (42 % for ILLIIM and 49% for RNYIAQVD). The main difference being the twist of the beta-sheets, according to the analysis ILLIIM is predominantly left twisted and RNYIAQVD is right twisted. The remaining secondary structure for both assemblies is predicted to be dominated by **1** turns as might be expected for this small amyloidogenic peptide (helical structures would be impossible for 6 and 8-mers). (see page 6, lines 44 page 7 line 8).

“The authors mentioned on page 4, “Simulations were run in explicit water (T!P3P37) using the ff15ipq forcefield38 and the pmemd time integrator39.” However, authors have not discussed the results of the simulations in the manuscript draft. Moreover, the details about the simulation length, box size, number of water molecules etc. should be included in the manuscript text.”

We have added the box size and number of water molecules, next to the statement of the time length of the simulation, in the methods section (page 4, lines 20-21). The results from the simulations which the authors expect to be of interest are the steric zipper structure, and its qualitative agreement (in the unit-cell defining region around q roughly equal to 1) with the WAXS from the solution ensemble, which are presented graphically in figure 4.

“The authors need to provide the full form of BeStSel in the manuscript text.”

The full name of the BestSel (Beta Structure Selection) program has now been provided along with complete

references to the papers that full describe the algorithms used (Pg 4, line 15).

Reviewer 3

This study extensively demonstrates that this novel candidate sequence is amyloidogenic by various analytical techniques. However, it has been known that partial peptides of proteins can be amyloidogenic.

In addition to lysozyme, which the authors describe in the main text, P8 (p53(250–257)) in the tumor suppressor p53 has been reported to be amyloidogenic <https://doi.org/10.1021/bi500825d>.

It is true that partial peptides of proteins are certainly amyloidogenic, with lysozyme and p53 being two examples (of which there are many others). However, generally these short peptides represent the amyloid forming core of a protein, which upon being provided the correct stimulus (i.e. seeding or partial denaturation) will drive amyloid formation in the full protein. This is shown in the above paper on p53 and also frequently with lysozyme. The ORF proteins studied here are very short proteins (ORF6 is 61 and ORF10 is 38 residues), therefore these amyloidogenic short peptides make up a relatively higher percentage of the full amino acid sequence than is the case with p53 or lysozyme (or indeed most proteins). Thus, it is even more likely that the full ORF6 and ORF10 proteins will maintain this amyloid forming ability.

“is there any evidence that fragmented peptides of ORF6 and ORF10 proteins exist in vivo?”

To the best of our knowledge there is no evidence of this, however this is not what we are suggesting. We are using these short peptides as model systems to predict the behavior of the full proteins. This allows us to perform biophysical and biochemical assays on these assemblies in a rapid manner, and screen a number of different peptides. We plan in follow-up studies (that are currently underway) to investigate if the full proteins identified from these fragments do indeed undergo self-assembly into amyloid nanostructures. These experiments will be done both in buffered solutions, and the proteins will be expressed in eukaryotic cells to determine if amyloid aggregation of ORF proteins occurs in the cytoplasm of neuronal cells.

“do full-length or in vivo processed ORF6 and ORF10 proteins form amyloid?”

This is a very important question, in discussions we were inspired to do this work by a similar question, do any of the proteins in SAR-CoV-2 form amyloids *in vivo*? Whilst the proteome of the virus is relatively small this still gave us around 40 proteins to choose from. This initial work has enabled us to narrow down our search from the full proteome to two lead candidates. We have now identified collaborators who work in virology and cell biology departments and work is underway to express these proteins in *E. coli* and in eukaryotic cells (to study their behavior in the cytoplasm). These experiments will answer the reviewers question however to do them thoroughly will take 9-12 months (including protein purification and repeating all of the biophysical and biochemical analysis performed on the peptides). Due to the potential relevance of this work, we suggest that these experiments on the full proteins should be the scope of a follow-up paper.

“do ILLIIM and RNYIAQVD cause amyloid aggregation of ORF6 and ORF10 proteins in their native forms (full-length or processed length) in vivo?”

This is an interesting question, and once we have setup the system of ORF6 and ORF10 expression in neuronal cell lines we will perform these seeding experiments similarly to the experiments elegantly performed in Ghosh et. al. with the p53 protein. Once again, we suggest that these in-depth cell biology experiments are beyond the scope of this manuscript, and will be carried out in a following work.

“this study should explain the rationale for not selecting D30YIINLIKNL.”

The authors agree that from a biophysical perspective DYIINLIKLN is a very interesting sequence, however the homology between ILLIIM and ILQINS, which the authors are familiar with as a disease-related amyloid fragment, made this shorter peptide irresistible as the first point of entry to the aggregation properties of the ORF6 peptide.

“does it form amyloid fibrils at physiological temperatures? in the M&M's description, the temperature is gradually lowered from 80°C, and it appears that fibrils are not formed unless the temperature is high.”

We have performed additional AFM experiments clearly showing that both assemblies do form aggregates at 37 °C, however the heterogeneity of structures formed was increased (see additional figure S3). Such heterogeneous distributions of assemblies are common in short peptides that have limited solubility, and the technique of adding warmed buffered solutions to the lyophilized powder and holding at this temperature for >2 hours is frequently used to maximize the solubility of the peptides, and by cooling slowly more homogenous distribution of assemblies are formed. Therefore, the elevated temperatures are certainly not required for amyloid formation, but were used to encourage more homogenous assembly, by providing as close to a fully solvated starting product as possible. We have now added an additional figure to the SI (Figure S3) showing the assemblies made at physiological temperatures, and added additional sentences to clarify these points. (see pg 3, lines 26-31 and pg 5, lines 25-34).

“!LL!M and RNY!AQVD can coexist in vivo. Do these peptides show cross-seeding?”

We have performed a number of additional AFM experiments to answer this question. Indeed when mixed in a 1:1 molar ratio the two peptides form a wide variety of polymorphic structures including ones that seem to be unique to the mixed samples (see Figure S8). These include 2D crystalline sheets with distinct edge morphologies presumably due to the molecular packing of these extended 2D crystals. This presents an intriguing hypothesis, that the much larger length-scale of flat assembly is made possible by the “resting” twists of ILLIIM and RNYIAQVD balancing out (left for ILLIIM and right for RNYIAQVD, see figure 3), however the concept of a sequence-dependent inherent chirality or “resting twist” of a peptide beta sheet is quite elusive and seems in our experience to have very sensitive dependence on many factors, so we can't be very confident that this detail of the co-assembly would translate to the *in vivo* case. See page 6, lines 16-24 for additional discussion).

“For AFM images, please describe the scanning range and the number of scanning lines in the y-direction, the number of pixels in the x-direction, the color scale and the corresponding z range in the Figure caption so that the reader can understand that spatial resolution.”

This has been done and the figure legend of figure 2 has been updated along with figures in the SI.

“for the histogram in Figure 2, please provide the equation for the fitting curves and the fitting parameters in the figure legend.”

Fitting was done using the proprietary Kernel smooth fitting function in Origin Pro, the figure legend has been updated to include the fitting parameters used.

“for the AFM analysis, the main text describes the thickness step of !LL!M as 12 nm, but in Figure S3 it looks clearly smaller than 12 nm. The appropriate explanation should be added.”

Indeed both peptides seem to show a highly polymorphic distribution of step heights, which is shown in our analysis. We have modified the text to make it clear that stacking of 2D beta sheet structures is likely the cause of this polymorphism in step height. (see pg 5, lines 48 to page 6 line 3).

“RNY!AQVD has twisted fibers in the upper left corner of Figure S4 and Figure 2i. This paper also explained that the distribution of layers is not a single step. RNYIAQVD can have structural polymorphisms, so please explain this point as well.”

We thank the reviewer for pointing out the polymorphic nature of the RNYIAQVD peptide, as a result of this we have performed additional AFM experiments and found that whilst the crystalline structures shown in figure 2 are

the dominant assembly, both peptides do indeed form a number of polymorphic structures (Figure 2, S3, S4 and S8). Additional imaging has revealed a number of fibrillar and crystalline polymorphs that were not previously observed (especially if the assembly is performed without the pre-heating step). We have added a significant amount of text discussing the polymorphs observed and how this might influence the levels of cytotoxicity seen by the different shaped assemblies (see page 5-6 and page 8-9). We believe that by correlating the polymorphism and morphological differences of the amyloid assemblies seen here with differing levels and trends in toxicity we have significantly strengthened this paper, and thank the reviewers for pointing out this oversight in the original manuscript.

Reviewer 4

"I find that the SAXS results do not contribute quantitatively to the findings. In addition, the SAXS experiments are conducted over a very limited q-range which is very moderate even for lab-based SAXS measurements. The q-range is from 0.02 to 0.25 Å⁻¹, which means that any quantitative information for the size of aggregates is impossible."

Whilst the SAXS plots do not add substantial additional novel information to the manuscript they do reinforce the conclusions made from analyzing the TEM and AFM images. Further, they provide evidence that the amyloid assemblies seen are not artifacts of drying or the solid support used for AFM or TEM. The SAXS clearly shows that the dominant polymorph is a flat 2D crystal, as shown by the presence of a scatter plot with q⁻² slope. See pg 6, lines 29-31.

The q-range investigated for the SAXS was 0.002 – 0.25 Å⁻¹, not 0.02 – 0.25 as quoted in the materials and methods, we sincerely apologize for this typographic error.

"For the two peptide sequences, a qualitative difference in the SAXS curve is observed and yet these differences are merely reported and not discussed further. The RNYIAQVD aggregates show a q^{"-4} dependence whereas the ILLIIM shows a q^{"-2} dependence. This difference is likely to arise from the ILLIIM aggregates having a broader lateral dimension as described. The q^{"-4} dependence could arise from a sharp interface between a scattering entity and its background, however, there is no discussion that this may be different for the two peptides."

We have now added additional discussion of the SAXS data pointing out that the clear q⁻² slope for the ILLIIM assemblies is likely a result of the assemblies having broader lateral dimensions thus backing up our imaging data. We have also added a brief discussion on the possible interpretations of the q⁻⁴ slope in the RNYIAQVD assemblies. (see page 6, lines 29-36).

"For the WAXS data, these were measured on the aggregates in solution, probably simultaneously as the SAXS data was collected. Wide-angle diffraction data from isolated needle crystals would of course provide much richer information."

We thank the reviewer for this suggestion but we suggest as the signal is already very strong in solution the bulk spectra of isolated needle crystals will only be more intense (revealing no additional data). This suggestion however has inspired us to attempt to perform synchrotron single crystal X-ray diffraction experiments to experimentally resolve the single crystal unit cell. However, fishing, isolating, shooting and resolving single crystals diffraction is a work of paramount importance, which furthermore requires a different beamline setup from the one used at the synchrotron where beamtime was granted. We take this remark as an inspiring comment for future work.

"The data show two diffraction peaks for ILLIIM which the authors identified as likely from a beta-sheet structure. The inter-sheet spacing was calculated to be 4.8 Å but typically this is often measured at 4.7 Å. No reason for this discrepancy is given."

As the reviewer points out inter-sheet spacing in extended beta sheets is most commonly reported at either 4.7 or 4.8 Å, and we report a distance of 4.6 Å. There are multiple reasons for this 0.1 Å discrepancy, it could well be due to inherent error of the detector used (0.1 Å is 30 times less than the bond length of a hydrogen atom, well

below the calibration error of any detector). Additionally depending on the precise amino acid structure and conditions this inter strands spacing can vary. Lomont et. al. (J. Phys. Chem. B., 2017, 121, 8935) used published amyloid molecular structures and observed that this intersheet spacing can vary as much as 0.45 Å. Finally, the CD data of the 1 mg mL⁻¹ ILLIIM structures suggests a complex mixture of relaxed and strained beta sheets, so it is perhaps expected that the average intersheet distance might deviate slightly from the standard value. We have added some additional sentences in the discussion to clarify this point. (see pg 7, lines 17-22).

“The cross beta-sheet peak at approx. 11 Å is sharp in comparison to the simulation. No explanation for this finding is given. Interpreting this as a beta-sheet structure is only possible with confidence when taking into account the CD data as well.”

Peak sharpness increases with the length scale of order in the structure, which unfortunately is limited by computer resources, so rounder peaks are to be expected when simulation is compared to a solution which may contain micron-sized aggregates. Even if an optimal size and shape to reproduce the solution scattering were found, it is likely that no single microcrystal could reproduce the scattering from a polydisperse solution having multiple potentially very distinct species. Thus, in our simulations we are limited to agreement between simulation and SAXS/WAXS on positions of the major peaks, illustrating beta sheet unit cell size and shape. Some words of clarification have been added (see page 7 lines 29-32).

“For RNYIAQVD a more complex WAXS pattern is shown without any indexing of the diffraction peaks. The authors comment that “RNYIAQVD has a number of well-defined Bragg peaks in between 0.3-0.8 Å⁻¹ which are consistent with a mixture of first and second order reflections corresponding to an amyloid-like 3D symmetry.” In the future, they should show how they arrive at this conclusion through indexing and referencing appropriately. A 3D molecular model may help in this assertion.”

It wasn't possible, regrettably, for the authors to solve a 3D model from the solution scattering, which is in general regarded as a much harder problem than the production of structures from single-crystal X-ray, for the very same reasons commented above. We can only assign with certainty one peak from RNYIAQVD, the one at $2\pi/4.6$ Å which it shares with ILLIIM and which is a strong signature of beta sheet. The group of peaks at 0.35 to 0.75 Å⁻¹ ($2\pi/18$ to $2\pi/8.3$ Å) to are very typical of combinations of the longer two axes of an octapeptide amyloid unit cell, however we couldn't find a combination of unit cell vectors (and lattice transform, and aggregate dimension, and aggregate twist) to generate them in-silico. We have added text to explain pg 7 lines 25-27.

“Differences in the amyloid structures for the two peptide sequences are shown from AFM, TEM and CD data alone. However, the paper fails to explore the significance of these differences upon the cytotoxicity. Indeed, the structural information provided by these physical characterization methods only serves to point to a possible amyloid structure at the peptide concentration used in these methods.”

We have added additional discussion referencing previous papers which show that cytotoxicity is sensitive to the size and shape of amyloid aggregates and not just their peptide sequence. We have now correlated aggregate morphology and free energy with cytotoxicity. Based on previous research we suggest that the short, wide ILLIIM assemblies are able to induce greater apoptosis than the longer, thinner RNYIAQVD assemblies, and that this is potentially due to a previously observed increase in cell membrane disrupting ability for shorter amyloid assemblies (backed up by the greater levels of apoptosis induced by the ILLIIM assemblies). In our opinion we have now provided a link between the differences in toxicity observed and the differing levels of cytotoxicity observed. (See pg 8 lines 46 to page 9 line 38).

In summary, we believe that the substantial alterations made to both the results and their interpretation have significantly improved the quality of this paper. We thank the editors and the reviewers for their detailed feedback, and hope that like-us you will now find this paper suitable for the interdisciplinary readership of Nature Communications.

Sincerely

REVIEWERS' COMMENTS

Reviewer #1 (Remarks to the Author):

The status of the revised manuscript has been improved after the revision. However, the vital question still lies unattended. Until and unless the presence of amyloids derived from SARS-CoV-2 proteins/peptides is established, the present study will have limited relevance. I agree with the authors that it is difficult, but I believe it is not impossible. In the absence of evidence of the existence of amyloid in in vivo systems, the argument of designing therapeutic strategies (solely based upon the amyloid theory) seems to be redundant.

A deliberation and input from clinicians handling the COVID-19 might be of great help. The nasal secretions from the COVID-19 patients might constitute a viable biological material where the presence of amyloids could be looked upon. Investigators must realize that the bottleneck of this study is to establish the presence of the SARS-CoV-2 proteins/peptide-derived amyloid in vivo. Otherwise, merely reporting the amyloid-forming ability of the ORF6 and ORF10 transcripts might not be of great importance.

On the other hand, the authors are suggested to use conformation-specific-antibodies for amyloids to authenticate the amyloid-forming ability of the protein/ peptide aggregates derived from ORF6 and ORF10 transcripts.

As mentioned previously, a graphical mechanistic model would be helpful in bringing more clarity to this observation.

The manuscript may be accepted after the careful illustration of the above comments.

Reviewer #2 (Remarks to the Author):

In this study, authors predicted the peptide fragments (ILLIIM and RNYIAQVD) in the SARS-CoV-2 proteome to be highly amyloidogenic using a bioinformatic scan. The study reported two peptides, ILLIIM and RNYIAQVD, from the open reading frame 6 (ORF6) and open reading frame 10 (ORF10) proteins, respectively, which rapidly self-assemble into amyloid aggregates and the amyloid assemblies displayed toxicity to a neuronal cell line. The authors suggested that the cytotoxic amyloid aggregates of SARS-CoV-2 proteins are responsible for the neurological symptoms observed in COVID-19 patients. In the revised manuscript, the authors have addressed the queries that were raised in the previous draft. The results of the study will be beneficial for the wider readership of the journal.

Response to Reviewers for Charnley *et. al.* Neurotoxic Amyloidogenic Peptides in the Proteome of SARS-COV2: Potential Implications for Neurological Symptoms in COVID-19

Detailed Response to Reviewers Comments (reviewers comments in italics, response in regular font, line numbers refer to the version of the manuscript with tracked changes).

Reviewer #1 (Remarks to the Author):

The status of the revised manuscript has been improved after the revision. However, the vital question still lies unattended. Until and unless the presence of amyloids derived from SARS-CoV-2 proteins/peptides is established, the present study will have limited relevance. I agree with the authors that it is difficult, but I believe it is not impossible. In the absence of evidence of the existence of amyloid in in vivo systems, the argument of designing therapeutic strategies (solely based upon the amyloid theory) seems to be redundant.

We thank the reviewer for their insightful comments, and we have toned down our discussion of designing therapeutic strategies based on the results of this manuscript. We agree perhaps this is slightly premature (see page 13, line 9 in the revised manuscript for edits).

Whilst it is true that we have not detected the presence of amyloids from SARS-CoV-2 in vivo, we have clearly demonstrated that the peptide fragments from ORF6 and ORF10 do indeed form readily form amyloid assemblies. The amyloid forming nature of these peptide has been proven through a multitude of analytical techniques regularly used to confirm the presence of amyloid assemblies, these include: Positive Thioflavin T (ThT) staining (a molecule that has long been used as a fluorescent marker for amyloid aggregates); the presence of extended β -sheet networks in the assemblies (by definition a nanofibril or crystal that is composed of extended β -sheet networks is considered an amyloid) as shown by both circular dichroism and wide angle X-ray scattering; the presence of twisted fibrils for RNYIAQVD assemblies (an almost ubiquitous feature of amyloid nanofibrils) as seen by atomic force microscopy and transmission electron microscopy; and the construction of an amyloid molecular unit cell that when relaxed via molecular dynamics simulations reproduces all the features of the experimental wide angle X-ray scattering curve. We would suggest that combined these results put it beyond doubt that these peptides readily form amyloid assemblies, and provide the necessary relevance to this paper.

A deliberation and input from clinicians handling the COVID-19 might be of great help. The nasal secretions from the COVID-19 patients might constitute a viable biological material where the presence of amyloids could be looked upon. Investigators must realize that the bottleneck of this study is to establish the presence of the SARS-CoV-2 proteins/peptide-derived amyloid in vivo. Otherwise, merely reporting the amyloid-forming ability of the ORF6 and ORF10 transcripts might not be of great importance.

We agree with the reviewer that confirming the presence of amyloid deposits in vivo is very important, and these

experiments should be performed by medical researchers and clinicians in dedicated labs with appropriate biological safety certificates allowing them to work with 'live' coronavirus samples. However, we do not have the capability to perform these experiments in a timely or safe manner. We thank the reviewer for the suggestion to get input from clinicians and as such have approached colleagues with links to clinical researchers working at the Walter and Eliza Hall Institute (WeHi) in Melbourne. These researchers have expertise in handling virally infected tissue (or indeed saliva samples) and access to the PC3 facilities at WeHi. However, these conversations are at a very early stage and it will take considerable time to fund, plan, execute and analyse the results of these experiments. Therefore, we ask that our initial findings are published sooner allowing other clinical researchers to plan their own experiments to answer some of the important outstanding questions about the in vivo presence of amyloids.

On the other hand, the authors are suggested to use conformation-specific-antibodies for amyloids to authenticate the amyloid-forming ability of the protein/peptide aggregates derived from ORF6 and ORF10 transcripts.

We thank the reviewer for this suggestion, and agree this will add further proof that the assemblies formed are indeed amyloid in nature (in addition to the proofs outlined above). We have performed experiments looking at the binding of a conformation specific antibody that binds to non-fibrillar amyloid oligomers (A11) and compared this to the binding of ThT (which fluorescently stains mature amyloid species). The results clearly show positively ThT stained assemblies further confirming the presence of amyloid aggregates. No A11 antibody binding was observed above the threshold set by the negative control (2⁰ antibody only), therefore we conclude whilst the assemblies formed are most definitely amyloid in nature (as shown by multiple techniques), under the conditions studied the presence of non-fibrillar oligomers could not be detected. Please see supplementary figure 9 and page 9, lines 1-11 for representative images and discussion of these experiments.

As mentioned previously, a graphical mechanistic model would be helpful in bringing more clarity to this observation.

We have added a graphical mechanistic model to the text (Figure 6) showing that the maturity of the fibrils (higher free energy twisted fibrils versus low free energy amyloid crystals) and their aspect ratio affects the extent of apoptosis in neurons co-cultured with these assemblies. We believe this figure now clearly summarises the results of this work. Once again we thank the reviewer for their suggestions.

Reviewer #2 (Remarks to the Author):

In this study, authors predicted the peptide fragments (ILLIIM and RNYIAQVD) in the SARS-CoV-2 proteome to be highly amyloidogenic using a bioinformatic scan. The study reported two peptides, ILLIIM and RNYIAQVD, from the open reading frame 6 (ORF6) and open reading frame 10 (ORF10) proteins, respectively, which rapidly self-assemble into amyloid aggregates and the amyloid assemblies displayed toxicity to a neuronal cell line. The authors suggested that the cytotoxic amyloid aggregates of SARS-CoV-2 proteins are responsible for the neurological symptoms observed in COVID-19 patients. In the revised manuscript, the authors have addressed the queries that were raised in the previous draft. The results of the study will be beneficial for the wider readership of the journal.

We thank the reviewer for their clear and concise summary of our work, and are very happy to hear that they believe the results of this study will be of interest to the readership of Nature Communication.